# HYDRA: TOWARDS TRANSFERABLE MULTI-TASK LEARNING ON TEMPORAL GRAPHS

## ABSTRACT

Real-world evolving networks are naturally modeled as temporal graphs (TGs), where capturing temporal dynamics is essential for predicting future graph properties that support downstream decision-making. Existing temporal graph methods have been developed primarily for single-task prediction, and little is known about their generalization across tasks or transfer to unseen networks. This leaves the challenge of multi-task graph property prediction in TGs largely open. We address this challenge by introducing Hydra, a novel architecture that integrates local connectivity features from temporal GNNs with a spectral learning module that captures global connectivity patterns. This design enables joint learning of local and global information under a multi-task objective. In multi-task classification, Hydra achieves an 8.9% relative gain in AUC over the strongest competitor. In multi-task regression, Hydra achieves competitive results in all three tasks, while obtaining the best results in two tasks with an 8.2% relative gain in MAE compared to the strongest baseline. Moreover, Hydra delivers these gains with a 22× reduction in training time compared to temporal transfer models. These results provide the first systematic evidence for effective multi-task transferable learning on temporal graphs. By delivering consistent top-ranked performance, Hydra highlights multi-task training on temporal graphs as a promising direction toward adaptable foundation models for temporal graphs.

## 1 INTRODUCTION

Temporal graph prediction addresses the challenge of forecasting the future state and dynamics of networks that evolve over time and possess an underlying relational structure. Unlike traditional graph analysis that assumes a static topology, temporal graphs are characterized by continuously changing nodes, edges, and features, representing dynamic real-world phenomena such as social network interactions (Wang et al., 2024), traffic flow (Sahili & Awad, 2023), and biological processes (Hosseinzadeh et al., 2022). The core difficulty lies in capturing both the complex spatial dependencies encoded in the graph structure and the temporal dynamics that governs its evolution (Kazemi et al., 2020). Effective solutions must therefore move beyond static graph representation learning to models that can jointly capture spatio-temporal patterns, enabling accurate prediction of future interactions, node properties, and system-wide events. This capability underpins applications such as disease outbreak detection (Senthilkumar et al., 2022), fraud prevention (Duan et al., 2024), recommendation systems (Tang et al., 2025), and transportation networks (Chen et al., 2022b). Despite progress in temporal graph learning, two fundamental challenges remain open.

**Multi-task Temporal Graph Prediction.** The first challenge is multi-task temporal graph prediction. Most existing methods are designed for single tasks, such as link prediction or node classification, but many real applications require forecasting multiple interdependent graph-level properties simultaneously. For example, one may need to predict both community evolution and connectivity growth. By learning these tasks jointly, models can exploit the shared spatio-temporal dynamics that shape them, yielding more robust generalization and a deeper understanding of system-wide behavior. Multi-task temporal graph prediction requires representations that transfer across heterogeneous graphs, remain stable under temporal drift, and still capture task-specific signals, which existing temporal GNNs and graph-level baselines cannot provide. Our framework overcomes these limits by using a shared trunk that fuses global structural dynamics with local temporal information to support robust multi-graph, multi-task prediction.

**Transferability Across Networks.** The second challenge is transferability across networks. In real-world dynamic settings, new networks constantly emerge, data may be sparse or incomplete, and training from scratch is costly. A practical solution is to pre-train temporal graph models on multiple observed networks and transfer them to unseen ones without retraining (Levie et al., 2021; Ruiz et al., 2023; Wang et al., 2025). This requires learning representations that capture generalizable temporal patterns while scaling effectively with data size and model capacity. Unlike static graphs, temporal graphs add the complexity of evolving structures and features, which makes transfer and scaling more difficult. Yet, no prior work examines how to build a representation trunk that captures cross-network temporal invariances while supporting multiple tasks through flexible lightweight heads. The tasks we study match real system needs: forecasting structural stability, connectivity growth, and temporal homophily, which are standard aggregate indicators used in risk analysis, mobility modeling, and community monitoring.

In this work, we address both challenges by introducing **Hydra**, a novel temporal multi-task graph neural network (GNN) designed for graph-level multi-property predictions with strong transferability across diverse networks. To capture global structural information, Hydra leverages graph Laplacian descriptors, while local node interactions and temporal dynamics are modeled through a Temporal Graph Neural Network (TGNN) equipped with a self-attention pooling layer. This trunk–head design separates a shared spectral–spatio–temporal backbone that captures cross-network, task-invariant dynamics from lightweight task-specific heads that flexibly support heterogeneous objectives. The resulting spectral and spatio-temporal representations are fused and passed through task-specific prediction heads (Figure 1). Trained end-to-end in a multi-task setting, Hydra achieves state-of-the-art performance on six benchmark tasks spanning classification and regression. Significantly, it generalizes across unseen networks, consistently outperforming models trained from scratch on those networks. We also analyze its scaling behavior, showing that performance improves as more training networks are incorporated. Our contributions are summarized as follows:

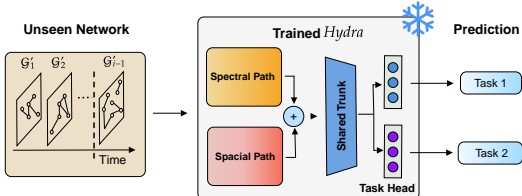

Figure 1: **Hydra overview.** Hydra is trained once on multiple networks and, without any additional training, can be directly applied to unseen networks for predicting multiple properties simultaneously.

- **First study on multi-task learning for temporal graphs.** We introduce the first model that jointly handles multiple temporal graph prediction tasks without retraining. Hydra explicitly addresses task interference and cross-network invariance and achieves state-of-the-art performance on graph-level temporal prediction.
- **Novel integration of TGNNs with spectral methods.** To effectively capture global information of the graph, we incorporate Laplacian descriptors into TGNNs for graph-level prediction, supported by strong empirical results. This fusion enables cross-network transfer by capturing invariances not represented in purely temporal or purely spectral models.
- **Comprehensive study of transfer and scaling.** We demonstrate that Hydra surpasses the strongest baselines, achieving an 8.9% relative gain on AUC in multi-task classification and an 8.2% relative gain on MAE in two regression tasks, while remaining competitive on the third without any task-specific training. We further show that transferability to unseen networks improves as the number of training datasets increases.

## 2 BACKGROUND AND RELATED WORK

**Temporal Graph Property Prediction.** Graph property prediction has been widely studied in the static setting, where various methods (Luo et al., 2025; Ying et al., 2021; Yang et al., 2023) have shown strong performance in tasks such as molecular property prediction (Sypetkowski et al., 2024; Wieder et al., 2020) and social network analysis (Chen et al., 2020). Building upon these advances, temporal graph property prediction has recently received increasing attention. One of the pioneer approaches to extend static models into temporal settings is GraphPulse (Shamsi et al., 2024) that combines recurrent architectures with topological data analysis tools to capture temporal patterns. However, existing work has largely been limited to training on a single network (Huang

et al., 2023b; 2024), with the exception of MiNT (Shamsi et al., 2025), which trains across multiple networks and demonstrates transferability to unseen networks during the inference phase. However, MiNT is restricted to a single classification task, and its computational cost is high due to the use of a hyperbolic backbone model. In this work, we go beyond this limitation by addressing multiple classification tasks as well as regression tasks in a scalable manner.

**Multi-Task Learning.** Multi-task learning has been extensively studied as a means to improve generalization by jointly optimizing related tasks and leveraging shared structural information (Caruana, 1997; Zhang & Yang, 2018; 2021). In the context of graph learning, a variety of approaches have been proposed that integrate learning strategies to enhance generalization, achieving strong performance across diverse datasets and downstream tasks (Jin et al., 2022; Ju et al., 2023; Klaser et al., 2024; Nassif et al., 2020). However, while these advances have yielded significant progress for static graphs, the application of multi-task learning to temporal graphs remains largely unexplored.

**Temporal Graph Transferability.** Recent advances in temporal graph learning have produced models tailored for tasks such as link prediction (Rossi et al., 2020; Huang et al., 2023a; Jin et al., 2023). While these methods achieve strong results on specific datasets, their applicability is often narrow: models trained for one task or on one network may fail to generalize to new settings. Unlike static graphs, temporal graphs introduce additional challenges for transfer, including evolving topologies, distribution shifts, and non-stationary dynamics. These factors make it difficult for a model to retain predictive power outside its training environment. Addressing transferability is important for real-world use cases where networks are constantly changing, data collection is costly, and retraining for every task or dataset is impractical (Wang et al., 2020; Huang et al., 2023b; Qi et al., 2023). A central open problem is to design models that capture generalizable temporal patterns so that knowledge gained from past networks or tasks can be reused in unseen ones (Pan et al., 2025; Liang et al., 2025). Developing such models would enable scalable, efficient, and continuous prediction in dynamic environments.

## 3 HYDRA: A MULTI-TASK, TRANSFER LEARNING MODEL

### 3.1 PROBLEM DEFINITION

We define a discrete time temporal graph as a sequence of snapshots $\mathcal{G} = \{\mathcal{G}_t = (V_t, E_t, X_t)\}_{t=1}^T$, where $V_t$ and $E_t$ denote the node and edge sets at time $t$, and $X_t$ represents associated node or edge features. In this work, we consider a collection of temporal graphs $\mathcal{D} = \{\mathcal{G}^{(i)}\}_{i=1}^N$, each evolving independently over time. For each temporal graph $\mathcal{G}^{(i)} = \{\mathcal{G}_t^{(i)}\}_{t=1}^{T_i}$ we associate task-specific labels $\{y_t^{(i,k)}\}_{k=1}^K$ for $K$ prediction tasks.

*Definition* 1 (Multi-Task Temporal Graph Property Prediction). The objective is to learn a single model

$$f_\theta : \left\{ \{\mathcal{G}_t^{(i)}\}_{t=1}^{T-1} \}_{i=1}^{|\mathcal{D}|} \right\} \mapsto \left\{ \left( \hat{y}_t^{(i,1)}, \ldots, \hat{y}_t^{(i,K)} \right) \right\}_{i=1}^{|\mathcal{D}|},$$

that ingests all snapshots of all graphs in $\mathcal{D}$ up to time $t - 1$ and jointly predicts $K$ graph-level properties at time $t$ for each graph. The parameters $\theta$ are optimized jointly across $\mathcal{D}$, and the trained model is required to generalize to temporal graphs not included in $\mathcal{D}$.

Hydra is evaluated on two categories of prediction tasks across multiple temporal graphs. In the **classification setting**, for $k \in \mathcal{K}_{\text{cls}}$, the objective is to predict binary outcomes $\hat{y}_t^{(i,k)} \in \{0, 1\}$ indicating the directional change of graph-level properties between consecutive snapshots. In the **regression setting**, for $k \in \mathcal{K}_{\text{reg}}$, the model predicts numerical values $\hat{y}_t^{(i,k)} \in \mathbb{R}$ that quantify the magnitude of graph-level properties at the next snapshot. In principle, this division allows Hydra to handle both discrete and continuous objectives within a single unified architecture.

Multi-task learning provides the foundation for this setting (Zhang & Yang, 2021; Ruder, 2017). Instead of training and maintaining a separate model for each task, it enables simultaneous prediction, allowing shared temporal representations to capture interdependencies between tasks (Crawshaw, 2020). Hydra builds on this foundation by unifying diverse temporal prediction tasks under a single architecture. Specifically, we focus on a *shared trunk* architecture with task-specific heads as they have proven effective in balancing information sharing with the avoidance of negative transfer

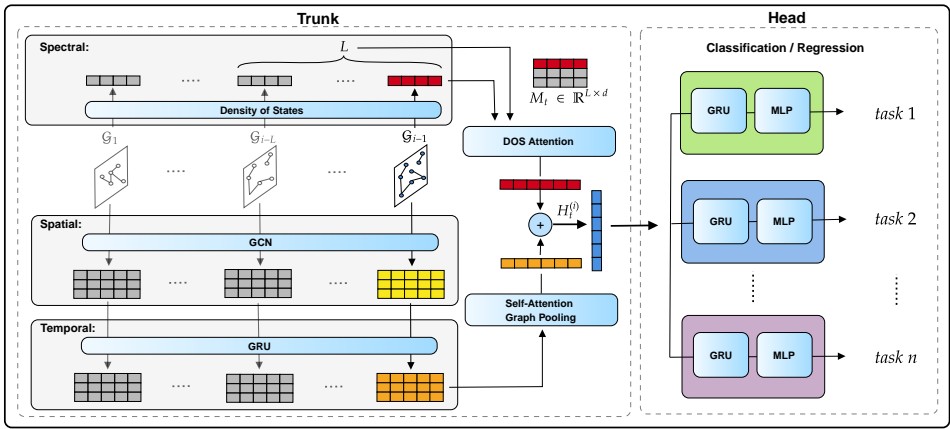

Figure 2: **Hydra framework.** The architecture integrates spatial and spectral paths with a unified multi-task design, enabling Hydra to learn transferable representations across diverse temporal graph tasks.

(Crawshaw, 2020). Learning tasks jointly allows the model to exploit shared temporal dynamics rather than treating each task in isolation. Figure 2 shows the Hydra framework.

## 3.2 HYDRA: SHARED TRUNK WITH TASK-SPECIFIC HEADS

The central design choice of Hydra is to couple a shared representation trunk with lightweight task-specific heads. We adopt this structure because the underlying challenges in temporal graph property prediction require representations that are both transferable across heterogeneous graphs and adaptable to multiple tasks. The trunk captures common temporal–structural patterns once, while task-specific heads provide minimal but sufficient specialization.

Hydra instantiates the predictor $f_\theta$ (Definition 1) with the shared trunk that produces embeddings $H_t^{(i)}$ for $\mathcal{G}_t^{(i)}$, followed by task-specific heads $\{g_{\psi_k}\}_{k=1}^K$ that map $H_t^{(i)}$ to predictions $\hat{y}_t^{(i,k)}$.

The trunk jointly leverages (i) spectral descriptors that summarize global connectivity patterns and (ii) spatial GNN-based encoders that model local temporal interactions. These components provide complementary global and local information while avoiding redundant representation learning.

As shown in Figure 2, the trunk $H_t^{(i)}$ consists of a spectral path, which encodes global descriptors, and a spatial path, which models local temporal dynamics. Their outputs are concatenated into the shared embedding $H_t^{(i)}$, which is then passed to task-specific decoder heads, $g_{\psi_k}{}_{k=1}^K$, each producing a prediction $\hat{y}_t^{(i,k)}$.

### 3.2.1 SPECTRAL PATH FOR GLOBAL CONNECTIVITY

Laplacian eigenvalues effectively encode global graph structural properties (Mohar et al., 1991). For example, in star, path, and fully-connected graphs, the distribution of the Laplacian eigenvalues is distinct and captures their unique topologies. However, computing all eigenvalues of the graph Laplacian matrix is highly expensive and only feasible for small graphs with thousands of nodes. Thus, it is important to efficiently approximate the spectrum of the Laplacian.

**Network Density of States.** To capture information from the Laplacian spectrum, we use the network density of states (DOS), the distribution of the eigenvalues of the graph Laplacian matrix (Dong et al., 2019), as follows:

*Definition* 2 (Density of States). Given a temporal graph snapshot $\mathcal{G}_t$ with adjacency matrix $\mathbf{A}$ and degree matrix $\mathbf{D}$, let the (unnormalized) Laplacian be $\mathbf{L} = \mathbf{D} - \mathbf{A}$. We use the normalized Laplacian $\mathbf{L}_{\text{sym}} = \mathbf{I} - \mathbf{D}^{-\frac{1}{2}}\mathbf{A}\mathbf{D}^{-\frac{1}{2}}$ with eigendecomposition $\mathbf{L}_{\text{sym}} = \mathbf{Q}\mathbf{\Lambda}\mathbf{Q}^\top$, where

$\mathbf{\Lambda} = \text{diag}(\lambda_1, \ldots, \lambda_{|V_t|})$ contains the eigenvalues. The *density of states* is then defined as a discrete distribution supported on the spectrum of $\mathbf{L}_{\text{sym}}$: $\mu(\lambda) = \frac{1}{|V_t|} \sum_{i=1}^{|V_t|} \delta(\lambda - \lambda_i)$

Essentially, $\mu(\lambda)$ measures the fraction of eigenvalues equal to $\lambda$. In practice, the range of the eigenvalues is discretized into multiple equal-sized intervals, and the number of eigenvalues within each interval is approximated. The Kernel Polynomial Method (Weiße et al., 2006) with Chebyshev expansions is used to efficiently approximate the spectrum.

**Spectral Memory.** To stabilize spectral signals and capture short-term trends, we maintain a fixed-length FIFO memory of recent DOS descriptors. At snapshot $t$, let $s_t \in \mathbb{R}^d$ denote the spectral feature (e.g., a $d{=}20$ DOS vector). For prediction at time $t$, the memory contains the last $L$ descriptors up to $t-1$: $M_t = [s_{t-L}; \ldots; s_{t-1}] \in \mathbb{R}^{L \times d}$. After prediction at $t$, the new descriptor $s_t$ is computed and rolled into the memory: $M_{t+1} = \text{roll}(M_t), \quad M_{t+1}[-1,:] = s_t$. This bounded memory reduces variance in non-stationary settings and preserves context across long temporal traces, typical of sparse graphs.

**Attention over Spectral Memory.** Rather than uniform averaging, we apply attention to emphasize historically similar or predictive patterns. With learnable projections $W_q, W_k, W_v \in \mathbb{R}^{d \times d_a}$, we compute $q_t = s_{t-1} W_q, \quad K_t = M_t W_k, \quad V_t = M_t W_v$. Attention weights and the attentive spectral embedding are $\alpha_t = \text{softmax}\left(\frac{q_t K_t^\top}{\sqrt{d_a}}\right), \quad z_t = \alpha_t V_t$. This mechanism highlights relevant history, suppresses noise, and captures structural–spectral motifs that recur across networks.

### 3.2.2 SPATIAL PATH FOR LOCAL CONNECTIVITY

We integrate spatial and temporal dynamics by combining a Graph Convolutional Network (GCN) (Kipf, 2016) with Gated Recurrent Units (GRUs) (Cho et al., 2014). Similar approaches have proven effective in temporal settings (Zhao et al., 2019). At snapshot $t$, a GCN encodes structural features $\mathbf{Z}_t = \text{GCN}(\mathbf{X}_t, \mathcal{E}_t)$, and a GRU updates node states with temporal memory $\mathbf{H}_t = \text{GRU}(\mathbf{Z}_t, \mathbf{H}_{t-1})$. The representation $\mathbf{H}_t$ is thus spatially grounded and temporally enriched.

**Pooling and Graph Readout.** We apply Self-Attention Graph Pooling (SAGPool) (Lee et al., 2019) to emphasize structurally important nodes. Node scores are computed as $s = \text{GCN}(\mathbf{H}_t, \mathcal{E}_t) W_s$, where $W_s \in \mathbb{R}^{d \times 1}$ projects node embeddings to scalar scores. The top-$k$ nodes are retained to form the pooled graph $(\mathcal{V}'_t, \mathcal{E}'_t)$ with embeddings $\mathbf{H}'_t$. We then compute a graph-level vector using mean pooling over the selected nodes: $g_t = \frac{1}{|\mathcal{V}'_t|} \sum_{v \in \mathcal{V}'_t} h_v, \quad h_v \in \mathbf{H}'_t$. SAGPool reduces noise from low-activity nodes and highlights structurally central actors, which is particularly relevant in sparse networks where activity is concentrated in a few key nodes (Shamsi et al., 2024).

## 3.3 PREDICTION HEAD

We fuse spatial and spectral signals to produce a joint temporal state. At each step, the shared hidden state of the prediction head $u_t = \text{GRU}([g_t \| z_t], u_{t-1})$, where $[g_t \| z_t]$ denotes concatenation of spatial and spectral features. The recurrent update captures cross-modal dependencies over time. Task-specific MLPs then map the shared state to predictions: $\hat{y}_t^{(k)} = \text{MLP}_k(u_t), \quad k = 1, \ldots, K$. This balances efficiency through a shared backbone with flexibility through task-specific heads.

## 3.4 END-TO-END TRAINING

`Hydra` supports both classification and regression tasks. While multi-task learning is the intended setting, we found that naïvely combining heterogeneous objectives degraded stability, consistent with prior reports (Yu et al., 2020; Sener & Koltun, 2018). A principled treatment of loss balancing is an open challenge and beyond our scope. We therefore train classification and regression groups separately, which ensures stable optimization while preserving `Hydra`'s transferability across unseen networks.

**Classification.** For each task $k \in \mathcal{T}_{\text{cls}}$, with labels $y_i \in \{0, 1\}$ and predictions $\hat{y}_i$,

$$\mathcal{L}_{\text{cls}}^{(k)} = -\tfrac{1}{N} \sum_{i=1}^{N} \left[ y_i \log \hat{y}_i + (1 - y_i) \log(1 - \hat{y}_i) \right], \quad \mathcal{L}_{\text{total}}^{\text{cls}} = \sum_{k \in \mathcal{T}_{\text{cls}}} \mathcal{L}_{\text{cls}}^{(k)}.$$

**Regression.** For each task $k \in \mathcal{T}_{\text{reg}}$, with labels $y_i$ and predictions $\hat{y}_i$,

$$\mathcal{L}_{\text{reg}}^{(k)} = \tfrac{1}{N} \sum_{i=1}^{N} (\hat{y}_i - y_i)^2, \quad \mathcal{L}_{\text{total}}^{\text{reg}} = \sum_{k \in \mathcal{T}_{\text{reg}}} \mathcal{L}_{\text{reg}}^{(k)}.$$

## 4 EXPERIMENTAL EVALUATION

### 4.1 TRAINING SETUP AND ALGORITHM

**Datasets.** We evaluate `Hydra` on the MiNT benchmark of 84 ERC-20 temporal transaction networks (Shamsi et al., 2025), randomly selecting 64 networks for training/validation and 20 held-out networks for zero-shot transfer evaluation. Within each network, snapshots are split 70%–15%–15% into training, validation, and test sets. A detailed dataset statistics table is shown in Appendix C. This setting enables us to evaluate whether `Hydra` can generalize across various network topologies and temporal patterns. During each epoch, networks are presented in random order, and historical embeddings are reset at the start of each network to avoid cross-network leakage. The shared trunk is optimized jointly, while all task-specific heads are updated under a multi-task loss with early stopping on validation performance. Models are trained for 100 to 250 epochs with an initial learning rate of $1.5 \times 10^{-3}$. Early stopping is applied based on validation AUC with a patience of 20 epochs and a tolerance of $5 \times 10^{-2}$. Binary Cross-Entropy loss is used for classification tasks, while Mean Squared Error is used for regression tasks. Optimization is performed using Adam (Kingma & Ba, 2015). Each task is evaluated on 20 unseen test networks with three independent runs, yielding 60 result points per task. We report results for each task separately, using ROC-AUC for classification objectives and Mean Absolute Error for regression objectives. We pre-train `Hydra` with 8, 16, and 32 networks to study the effect of scale on transferability. We cap the setting at 32 networks to allow multiple, independently sampled `Hydra`-32 models for analyzing the impact of data selection. Unless otherwise specified, all reported results refer to `Hydra`-32, as this configuration consistently delivers the strongest performance.

**Tasks.** We train `Hydra` in two multi-task settings: a classification model with three classification heads and a regression model with three regression heads. The classification tasks jointly predict whether the Largest Connected Component (LCC), the number of nodes, or the number of edges will grow or shrink. For the regression tasks, we estimate the number of influential nodes (i.e., nodes with degree greater than five), the number of new nodes, and the number of edges. Detailed task definitions and their significance are provided in Appendix E.

**Models.** We evaluate `Hydra` in both multi-task classification and regression settings against state-of-the-art temporal graph learning baselines, including HTGN (Yang et al., 2021), GC-LSTM (Chen et al., 2022a), EvolveGCN (Pareja et al., 2020), GraphPulse (Shamsi et al., 2024), ROLAND (You et al., 2022), TGCN (Zhao et al., 2020), and WinGNN (Zhu et al., 2023). For classification, we additionally include MiNT (Shamsi et al., 2025), the only existing framework explicitly designed for transferability across temporal graphs. MiNT is excluded from regression since it does not support regression objectives. We distinguish between two categories of approaches: (i) *single models*, trained separately on each dataset, and (ii) *transferable models*, trained once and applied directly to unseen networks without retraining. `Hydra` belongs to the latter category, enabling true zero-shot transfer. Our code is available at: https://anonymous.4open.science/r/HydraTG/

**Compute Resources.** Experiments were run on a dedicated cluster compute node equipped with an NVIDIA H100 GPU (80 GB), dual 64-core AMD EPYC CPUs, and 512 GB RAM.

### 4.2 CLASSIFICATION AND REGRESSION RESULTS

Table 1 presents the aggregate classification results across three tasks: LCC Growth/Shrinkage, Node Growth/Shrinkage, and Edge Growth/Shrinkage. For each task, we report three metrics: (*1st*)

Table 1: Classification task results. Across LLC-GS, Node-GS, and Edge-G/S classification, `Hydra` consistently outperforms both specifically trained and transferable baselines.

| | LLC-G/S | | | Node-G/S | | | Edge-G/S | | |
|---|---|---|---|---|---|---|---|---|---|
| | 1st↑ | Rank↓ | AUC↑ | 1st↑ | Rank↓ | AUC↑ | 1st↑ | Rank↓ | AUC↑ |
| **Single Models** | | | | | | | | | |
| HTGN | 2 | 4.40 | 0.648 | 0 | 3.65 | 0.615 | 1 | 4.85 | 0.684 |
| GC-LSTM | 0 | 6.00 | 0.572 | 0 | 6.65 | 0.489 | 1 | 5.80 | 0.609 |
| EvolveGCN | 0 | 6.30 | 0.585 | 0 | 7.00 | 0.487 | 0 | 6.10 | 0.626 |
| GraphPulse | 3 | 3.50 | 0.686 | 2 | 2.95 | 0.652 | 6 | 3.80 | 0.712 |
| ROLAND | 0 | 6.95 | 0.556 | 0 | 6.70 | 0.480 | 0 | 6.30 | 0.542 |
| TGCN | 1 | 6.10 | 0.583 | 0 | 6.15 | 0.530 | 0 | 5.60 | 0.633 |
| WinGNN | 0 | 6.10 | 0.585 | 0 | 6.20 | 0.528 | 0 | 5.55 | 0.642 |
| **Transfer Models** | | | | | | | | | |
| MiNT | 1 | 3.95 | 0.672 | 3 | 3.55 | 0.616 | 4 | 3.30 | 0.727 |
| `Hydra` (ours) | **13** | **1.70** | **0.759** | **15** | **1.95** | **0.734** | **8** | **2.80** | **0.753** |

*Arrows indicate directionality:* ↑ higher is better, ↓ lower is better.

Table 2: Regression task results. Across New Node Count, Influential Node Count, and Edge Count regression, `Hydra` achieves the best performance on two tasks and is competitive on the third, all without additional training.

| | New Node Count | | | Influential Node Count | | | Edge Count | | |
|---|---|---|---|---|---|---|---|---|---|
| | 1st↑ | Rank↓ | MAE↓ | 1st↑ | Rank↓ | MAE↓ | 1st↑ | Rank↓ | MAE↓ |
| **Single Models** | | | | | | | | | |
| HTGN | 4 | 3.45 | 0.039 | 4 | **3.52** | **0.050** | 2 | 3.95 | 0.049 |
| TGCN | 1 | 4.10 | 0.042 | 1 | 4.85 | 0.059 | 1 | 4.60 | 0.054 |
| GCLSTM | 2 | 4.00 | 0.043 | **5** | 3.80 | 0.053 | 4 | 4.58 | 0.057 |
| ROLAND | 3 | 4.35 | 0.052 | 2 | 4.55 | 0.057 | 3 | 4.58 | 0.050 |
| EGCN | 0 | 3.90 | 0.040 | 0 | 4.20 | 0.052 | 1 | 5.30 | 0.054 |
| GraphPulse | 1 | 5.75 | 0.071 | 1 | 5.90 | 0.078 | 2 | 5.72 | 0.073 |
| WinGNN | 4 | 4.05 | 0.054 | 4 | 4.28 | 0.059 | 5 | 3.93 | 0.057 |
| **Transfer Model** | | | | | | | | | |
| `Hydra` (ours) | **8** | **2.50** | **0.035** | 5 | 4.90 | 0.056 | **6** | **3.35** | **0.046** |

*Arrows indicate directionality:* ↑ higher is better, ↓ lower is better.

the number of datasets where the method achieved first place, (*Rank*) the average ranking across datasets, and (*AUC*) the average area under the ROC curve (metrics formulations are provided in appendix F). `Hydra` consistently achieves the strongest performance across all metrics, outperforming both state-of-the-art specifically trained models and the transferable baseline. Detailed per-dataset classification outcomes are provided in Appendix I.

Table 2 summarizes the regression results across New Node Count, Influential Node Count, and Edge Count Regression. For each task, we report three metrics: (*1st*) the number of datasets where the method achieved first place, (*Rank*) the average ranking across datasets, and (*MAE*) the average mean absolute error. `Hydra` is the best model on two of the three tasks and remains very close to the best baseline on Influential Node Count, demonstrating strong overall transferability for regression objectives. Full results by dataset are included in Appendix I.

`Hydra` produces the most top-ranked results in both classification and regression settings, outperforming all baselines in five out of six tasks. To keep the main paper concise, we report summarized results in the main text and provide extended per-dataset outcomes in Appendix I. Even in the one regression task where `Hydra` isn't the leader, it stays competitive with the strongest baseline. Notably, `Hydra` is always tested on unseen networks, while

Table 3: Meta ablation summary for Edge G/S classification. Average AUC and #1st-Place counts are reported with and without the corresponding component. Per-network results are given in Appendix Tables 6 and 7.

| Ablation | Avg. AUC | | #1st-Place | |
|---|---|---|---|---|
| | w/o | w/ | w/o | w/ |
| Density of States | 0.697 | **0.719** | 6 | **14** |
| Attention-based Pooling | 0.719 | **0.753** | 3 | **17** |

single models specifically trained for a task are evaluated on the test portion of the same networks they were trained on. Despite this tougher setting, `Hydra` consistently delivers better or comparable results, making it a reliable and adaptable framework for temporal graph learning.

**Ablation Studies.** We assess the impact of Hydra's key components by disabling the DOS module and the pooling layer. As summarized in Table 3, DOS improves performance from an average AUC of 0.697 to 0.719, raising the number of best-performing datasets from 6 to 14 and lowering the mean rank from 1.70 to 1.30. Pooling yields an even larger effect, increasing average AUC from 0.719 to 0.753 and improving the mean rank from 1.85 to 1.15, with gains on 17 out of 20 datasets. These results confirm that DOS provides complementary global spectral information, while pooling enhances the spatial path by emphasizing structurally important nodes, together strengthening Hydra's generalization ability.

### 4.3 HYDRA GENERALIZATION AND EFFICIENCY

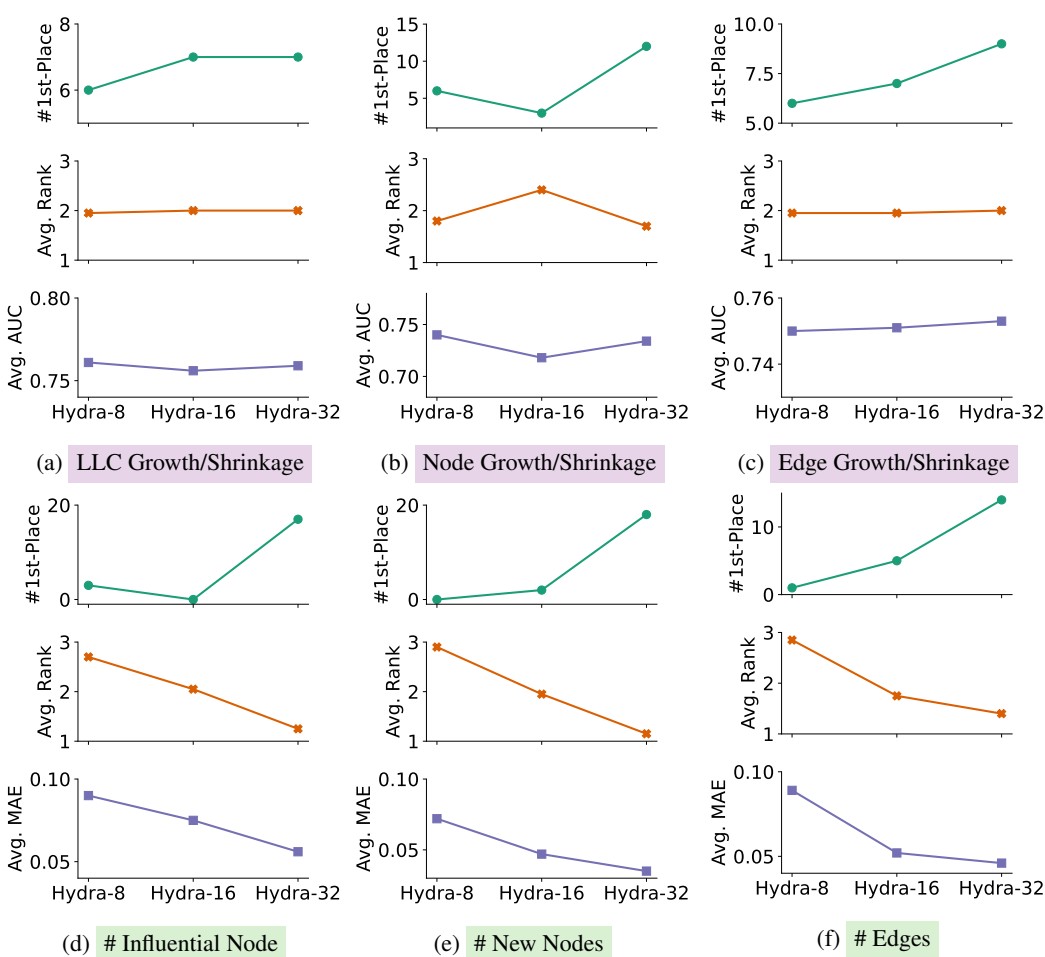

(a) LLC Growth/Shrinkage  (b) Node Growth/Shrinkage  (c) Edge Growth/Shrinkage

(d) # Influential Node  (e) # New Nodes  (f) # Edges

Figure 3: Impact of scaling the number of training networks for Hydra (from 8 to 32) on classification and regression tasks. A clear scaling trend is observed in most cases, with **Hydra-32** consistently delivering the best performance across all tasks. Evaluation measures are $\#1^{st}$-*Place*↑ (higher is better), *Avg. Rank*↓ (lower is better), *Avg. AUC*↑ (higher is better), and *Avg. MAE*↓ (lower is better).

In this section, we analyze how the number of networks used during the pre-training of Hydra affects its performance in predicting the characteristics of unseen networks at inference time. The results are presented in Fig. 3.

For classification tasks, we observe that increasing the number of training networks generally leads to a higher $\#1^{st}$-*Place*. This metric is a strong indicator of performance, as it reflects the number of unseen networks on which the model achieves the best results at inference time. In contrast, the correlation between the number of pre-training networks and metrics such as Avg. Rank or Avg. AUC is weaker, with only negligible differences across settings. Notably, Hydra-32 consistently

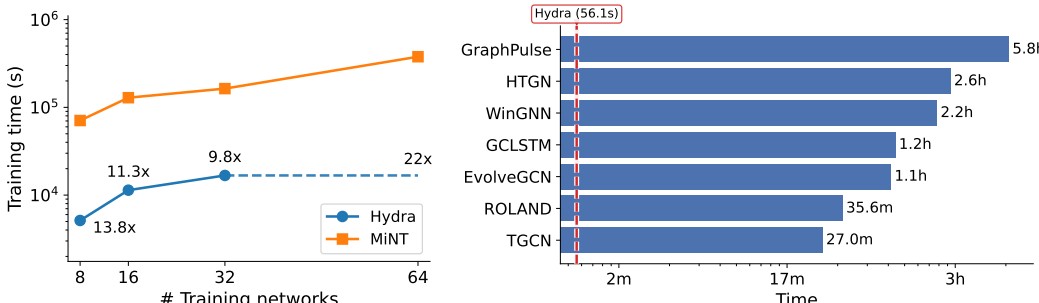

(a) **Total training cost for three classification tasks.** Hydra trains once for all tasks jointly, while MiNT must be retrained separately for each task. The dashed line indicates that MiNT is evaluated at 64 networks, while Hydra's best result occurs at 32

(b) **Inference cost on TRAC dataset.** Hydra performs a single zero-shot classification pass for all three tasks jointly, whereas single-network baselines require full training before they can be used for inference, even on a single task (LCC–G/S here); reported times reflect the cost of inference on an unseen network using Hydra versus single models.

Figure 4: **Training and inference efficiency of Hydra.** (a) Hydra reduces training cost by avoiding retraining. (b) Hydra drastically lowers inference time by requiring only one forward pass.

achieves the best or on-par performance across all classification benchmarks. For regression tasks, we observe a clearer scaling trend: performance steadily improves from Hydra-8 to Hydra-32 across all indicators. Specifically, $\#1^{st}$-*Place* increases with scale, while both Avg. Rank and Avg. MAE decrease, indicating better predictive performance.

Overall, Hydra-32 consistently achieves the strongest results, with the highest $\#1^{st}$-*Place*, the lowest Avg. Rank, and the best average performance metric (i.e., highest Avg. AUC for classification and lowest Avg. MAE for regression). Given its consistent superiority across all six tasks, we adopt **Hydra-32** as our primary model and center our subsequent evaluation on its performance. For completeness, Table 8 in Appendix H reports the performance of all Hydra variants.

**Computation and Time Efficiency.** The per-snapshot complexity of Hydra reduces to a scalable $\mathcal{O}(N \cdot m + n \log n)$, where $m$ is the number of edges, $n$ is the number of nodes, and $N$ is the number of Chebyshev moments used in the DOS module (see Appendix B for details). Because MiNT must be retrained separately for each task, training three tasks requires about three full runs on 64 networks. In contrast, Hydra trains all tasks jointly in a single run on 32 networks and achieves a 22× speedup, completing all three tasks in about 16,700 seconds compared to more than 370,000 seconds for MiNT (Figure 4a). This efficiency makes Hydra far more practical and scalable for multi-task training and inference.

Hydra inference time also shows a significant gain. Hydra requires only a single zero-shot inference pass to predict all three tasks on an unseen network, completing end-to-end evaluation on the largest test network (TRAC) in about 56 seconds (see Figure 4b). Single-network baselines, in contrast, require retraining from scratch for every dataset and task before they can be used for inference, with training times ranging from several thousand seconds to over 9,000 seconds even for a single task. Extending these models to multiple tasks or multiple networks multiplies the cost linearly, making them infeasible at scale. Hydra 's design eliminates the need for repeated training and heavy preprocessing, providing unique efficiency in both training and inference.

## 5 IMPACT OF MULTI-TASK TRAINING

To assess how multi-task learning influences the quality of the shared temporal representation in Hydra, we conducted an experiment designed to reveal the effect of adding new predictive objectives during training. We choose edge count prediction as the reference task. By observing how its performance shifts as additional tasks are introduced, we can directly evaluate how multi-task supervision shapes the learned representation. We trained three variants of the model: a single task version using only edge count, a two-task version trained on edge count and new node count, and

the full three-task configuration of `Hydra`. This setup allows a clear examination of how additional tasks contribute to the learning process.

The results in Table 4 reveal a consistent trend. As more tasks are incorporated into training, performance on the reference task improves across the majority of datasets. The two-task model surpasses the single-task model in the number of best-performing datasets, and the full three-task version achieves the best performance across most datasets and the strongest overall average, with 12 out of 20 datasets achieving the best performance. These gains arise because additional tasks introduce complementary supervisory signals that encourage the shared trunk to learn richer temporal and structural patterns. Rather than creating interference, this enhances the quality of the representation, demonstrating that `Hydra` effectively leverages cross-task information to strengthen performance on individual predictive objectives.

Table 4: MAE of edge count prediction under one, two, and three task Hydra configurations, reported with standard deviations.

| Dataset | Hydra (One Task) | Hydra (Two Tasks) | Hydra (Three Tasks) |
|---|---|---|---|
| MIR | $0.057 \pm 0.004$ | $0.034 \pm 0.002$ | $\mathbf{0.016} \pm 0.005$ |
| DOGE2.0 | $0.169 \pm 0.016$ | $\mathbf{0.150} \pm 0.015$ | $0.187 \pm 0.008$ |
| MUTE | $0.053 \pm 0.020$ | $0.067 \pm 0.011$ | $\mathbf{0.021} \pm 0.022$ |
| EVERMOON | $0.029 \pm 0.021$ | $0.042 \pm 0.010$ | $\mathbf{0.017} \pm 0.000$ |
| DERC | $0.024 \pm 0.016$ | $0.035 \pm 0.010$ | $\mathbf{0.021} \pm 0.003$ |
| ADX | $\mathbf{0.023} \pm 0.010$ | $0.027 \pm 0.006$ | $0.025 \pm 0.008$ |
| HOICHI | $0.071 \pm 0.006$ | $0.068 \pm 0.011$ | $\mathbf{0.027} \pm 0.020$ |
| SDEX | $0.060 \pm 0.023$ | $\mathbf{0.046} \pm 0.007$ | $0.095 \pm 0.043$ |
| BAG | $0.061 \pm 0.018$ | $0.067 \pm 0.020$ | $\mathbf{0.041} \pm 0.020$ |
| XCN | $0.037 \pm 0.017$ | $\mathbf{0.026} \pm 0.010$ | $0.062 \pm 0.015$ |
| ETH2x-FLI | $0.030 \pm 0.013$ | $0.038 \pm 0.007$ | $\mathbf{0.023} \pm 0.001$ |
| stkAAVE | $0.026 \pm 0.008$ | $\mathbf{0.021} \pm 0.003$ | $0.051 \pm 0.022$ |
| GLM | $0.092 \pm 0.009$ | $\mathbf{0.083} \pm 0.017$ | $0.087 \pm 0.008$ |
| QOM | $0.025 \pm 0.003$ | $\mathbf{0.023} \pm 0.005$ | $0.036 \pm 0.013$ |
| WOJAK | $0.025 \pm 0.023$ | $0.039 \pm 0.017$ | $\mathbf{0.010} \pm 0.004$ |
| DINO | $0.031 \pm 0.018$ | $0.041 \pm 0.010$ | $\mathbf{0.017} \pm 0.007$ |
| Metis | $0.047 \pm 0.014$ | $0.054 \pm 0.016$ | $\mathbf{0.034} \pm 0.010$ |
| REPv2 | $0.103 \pm 0.002$ | $\mathbf{0.102} \pm 0.003$ | $0.111 \pm 0.008$ |
| TRAC | $0.029 \pm 0.016$ | $0.039 \pm 0.012$ | $\mathbf{0.021} \pm 0.001$ |
| BEPRO | $0.031 \pm 0.025$ | $0.046 \pm 0.012$ | $\mathbf{0.011} \pm 0.008$ |
| **Average** | 0.051 | 0.052 | **0.046** |
| **Best Count** | 1 | 7 | **12** |

## 6 CONCLUSION

We have introduced `Hydra`, a novel temporal graph-level multi-task and multi-network model designed to address the complexities of dynamic network analysis. `Hydra` combines a spatial path, which captures local connectivity through temporal GNNs, along with a spectral path that attends to global structural patterns, forming a new architectural design for temporal graph learning. This unified framework enables `Hydra` to handle diverse prediction tasks simultaneously while transferring effectively to unseen networks. Empirically, `Hydra` consistently outperforms state-of-the-art baselines across benchmarks, achieving strong performance without requiring additional training on target networks. These results highlight `Hydra` as the first architecture to bring together spectral and spatial pathways for transferable, multi-task learning on temporal graphs. Looking ahead, this work opens up a promising research direction toward more generalizable temporal models that can support a wide range of tasks.

## REPRODUCIBILITY STATEMENT

We provide an anonymized Git at https://anonymous.4open.science/r/HydraTG/, which contains the full implementation of our models and experimental setup to ensure reproducibility. The code is also included in supplementary materials. Experimental results are reported as the mean and standard deviation across different random seeds, and the hyperparameters used are detailed in Section 4.1.

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

# Appendix

## A  LLM USAGE

We used LLMs to refine the writing of this paper. All ideas, content, and results are entirely our own; the LLM's role was limited to enhancing clarity, grammar, style, and LaTeX formatting.

## B  COMPUTATIONAL COMPLEXITY OF HYDRA

For a snapshot $G_t = (V_t, E_t)$ with $n = |V_t|$ nodes and $m = |E_t|$ edges, the per-snapshot complexity of Hydra is

$$\mathcal{O}\big(m \cdot d + n \cdot d^2 + n \cdot \log n + N \cdot m + K \cdot d\big),$$

where $d$ is the hidden dimension, $K$ is the number of task-specific heads, and $N$ is the number of Chebyshev moments used in the kernel polynomial method approximation of the DOS. The term $m \cdot d$ arises from spatial message passing, since each edge propagates a $d$-dimensional embedding. The term $n \cdot d^2$ comes from recurrent or attention-based updates. Pooling with top-$k$ selection contributes $n \cdot \log n$, reflecting the computation of attention scores for all nodes followed by partial sorting. The spectral DOS module adds $N \cdot m$, as each moment requires one sparse matrix–vector multiplication. Finally, the cost of task-specific heads is $K \cdot d$. Over $N$ networks with $T$ snapshots each, the overall training cost scales as

$$\mathcal{O}\big(N \cdot T \left[ m \cdot (d + N) + n \cdot d^2 + n \cdot \log n + K \cdot d \right]\big).$$

In our implementation, the spatial hidden dimension is fixed to 16, the DOS descriptor dimension is 20, and we train with $K = 3$ task-specific heads. Substituting these values, the per-snapshot complexity of Hydra becomes

$$\mathcal{O}\big(16 \cdot m + 256 \cdot n + n \log n + N \cdot m + 108\big),$$

where $m = |E_t|$ and $n = |V_t|$. The terms correspond to spatial message passing ($16 \cdot m$), recurrent/attention updates ($256 \cdot n$), pooling with top-$k$ selection ($n \log n$), DOS spectral approximation with $N$ Chebyshev moments ($N \cdot m$), and three task heads on a 36-dimensional joint embedding (108). For each snapshot, the overall training complexity scales as

$$\mathcal{O}\big(((16 + N)\, m + 256\, n + n \log n)\big).$$

In practice, with hidden size and Chebyshev moment parameters fixed, the per-snapshot complexity of Hydra reduces to $\mathcal{O}(N \cdot m + n \log n)$, dominated by sparse matrix–vector multiplications in the DOS module.

## C  DATASET

We evaluate Hydra on temporal transaction networks from the MiNT benchmark (Shamsi et al., 2025), the first large-scale dataset designed for training and evaluating temporal graph models across multiple heterogeneous networks. MiNT contains 84 ERC-20 token transaction graphs collected from the Ethereum blockchain between 2017 and 2023, spanning more than six years of activity. The biggest MiNT token network contains $128,159$ unique addresses and $554,705$ transactions, while the smallest token network has $1,454$ nodes. Each network is represented as a sequence of weekly snapshots, where nodes correspond to wallet addresses and edges represent token transfers. Across their full duration, networks range from tens of thousands to over 100K unique nodes and up to several million edges, reflecting the scale of real-world token ecosystems.

A distinctive feature of MiNT is its strong inductiveness: new addresses and transactions appear continuously, making prediction inherently open-world. This reflects the real behavior of blockchain networks, where adoption cycles, liquidity shocks, and market events constantly introduce novel participants and structural patterns. The long temporal duration further amplifies this effect, as networks exhibit bursts of rapid growth, periods of stagnation, and sudden fragmentation or collapse. Such novelty and surprise factors make MiNT particularly challenging, as models must generalize across highly dynamic trajectories rather than stationary or repetitive patterns.

Table 5: All token networks' statistics.

| Token | #Node | #Transaction | #Snapshots (days) | Growth rate | Novelty | Surprise | Token | #Node | #Transaction | #Snapshots (days) | Growth rate | Novelty | Surprise |
|---|---|---|---|---|---|---|---|---|---|---|---|---|---|
| ARC | 11325 | 70968 | 606 | 0.43 | 0.32 | 0.88 | Metis | 52586 | 343141 | 907 | 0.44 | 0.48 | 0.89 |
| CELR | 65350 | 235807 | 1691 | 0.49 | 0.56 | 0.96 | cDAI | 52753 | 358050 | 1437 | 0.45 | 0.46 | 0.9 |
| CMT | 86895 | 205961 | 309 | 0.45 | 0.72 | 0.92 | BITCOIN | 34051 | 347054 | 178 | 0.48 | 0.39 | 0.63 |
| DRGN | 113453 | 341849 | 2164 | 0.44 | 0.57 | 0.97 | INJ | 60472 | 312822 | 1113 | 0.46 | 0.52 | 0.98 |
| GHST | 35156 | 180955 | 1146 | 0.43 | 0.51 | 0.93 | MIM | 23038 | 269366 | 885 | 0.44 | 0.4 | 0.89 |
| INU | 8556 | 66315 | 154 | 0.27 | 0.41 | 0.59 | GLM | 53385 | 234912 | 1080 | 0.5 | 0.53 | 0.96 |
| IOTX | 63079 | 288469 | 1993 | 0.45 | 0.56 | 0.99 | Mog | 14590 | 240680 | 107 | 0.37 | 0.38 | 0.55 |
| QSP | 117977 | 299671 | 2178 | 0.45 | 0.67 | 0.99 | DPI | 40627 | 234246 | 1150 | 0.49 | 0.5 | 0.86 |
| REP | 83282 | 224843 | 346 | 0.46 | 0.69 | 0.96 | LINA | 45342 | 227147 | 1144 | 0.45 | 0.46 | 0.95 |
| RFD | 23208 | 173695 | 169 | 0.3 | 0.39 | 0.6 | Yf-DAI | 22466 | 226875 | 1158 | 0.42 | 0.31 | 0.87 |
| TNT | 88247 | 316352 | 1216 | 0.43 | 0.55 | 0.93 | BOB | 42806 | 212099 | 199 | 0.35 | 0.48 | 0.73 |
| TRAC | 71667 | 299181 | 2110 | 0.46 | 0.54 | 0.97 | RGT | 35277 | 211932 | 1110 | 0.44 | 0.46 | 0.98 |
| RLB | 28033 | 240291 | 129 | 0.43 | 0.49 | 0.76 | TVK | 42539 | 208082 | 1062 | 0.41 | 0.48 | 0.93 |
| steCRV | 19079 | 211538 | 1033 | 0.45 | 0.53 | 0.9 | RSR | 50645 | 205906 | 659 | 0.47 | 0.62 | 0.91 |
| ALBT | 63042 | 434881 | 1152 | 0.43 | 0.44 | 0.89 | WOJAK | 34341 | 198653 | 201 | 0.37 | 0.48 | 0.73 |
| POLS | 128159 | 554705 | 1132 | 0.45 | 0.61 | 0.94 | ANT | 36517 | 200262 | 1107 | 0.47 | 0.46 | 0.93 |
| SWAP | 69230 | 509769 | 1213 | 0.46 | 0.45 | 0.79 | LADYS | 37486 | 192176 | 181 | 0.37 | 0.52 | 0.79 |
| SUPER | 83299 | 502030 | 986 | 0.47 | 0.46 | 0.85 | ETH2x-FLI | 11008 | 199088 | 965 | 0.47 | 0.28 | 0.84 |
| RARI | 87186 | 502960 | 1207 | 0.43 | 0.47 | 0.91 | TURBO | 38638 | 189048 | 189 | 0.33 | 0.48 | 0.72 |
| KP3R | 39323 | 493258 | 1102 | 0.43 | 0.33 | 0.88 | REPv2 | 39061 | 191367 | 1194 | 0.48 | 0.5 | 0.97 |
| MIR | 79984 | 444998 | 1066 | 0.45 | 0.43 | 0.92 | NOIA | 29798 | 185528 | 1133 | 0.46 | 0.37 | 0.7 |
| aUSDC | 23742 | 475680 | 1067 | 0.46 | 0.4 | 0.73 | 0x0 | 21531 | 182430 | 283 | 0.51 | 0.46 | 0.81 |
| LUSD | 25852 | 430473 | 943 | 0.48 | 0.36 | 0.87 | PSYOP | 25450 | 168896 | 169 | 0.32 | 0.39 | 0.59 |
| PICKLE | 28498 | 430262 | 1149 | 0.48 | 0.34 | 0.69 | ShibDoge | 40023 | 134697 | 680 | 0.43 | 0.53 | 0.8 |
| DODO | 47046 | 390443 | 1131 | 0.47 | 0.45 | 0.91 | ADX | 14567 | 123755 | 1188 | 0.44 | 0.4 | 0.91 |
| YFII | 43964 | 391984 | 1196 | 0.44 | 0.44 | 0.96 | BAG | 11860 | 122634 | 298 | 0.31 | 0.44 | 0.87 |
| STARL | 71590 | 369913 | 856 | 0.46 | 0.48 | 0.86 | QOM | 21757 | 118292 | 598 | 0.46 | 0.41 | 0.81 |
| LQTY | 34687 | 374230 | 943 | 0.45 | 0.34 | 0.91 | BEPRO | 26521 | 120261 | 1132 | 0.46 | 0.48 | 0.87 |
| FEG | 118294 | 367584 | 1007 | 0.4 | 0.62 | 0.92 | AIOZ | 29231 | 119926 | 947 | 0.43 | 0.49 | 0.89 |
| AUDIO | 91218 | 362685 | 1108 | 0.45 | 0.58 | 0.95 | PRE | 40476 | 118625 | 1113 | 0.5 | 0.55 | 0.86 |
| OHM | 45728 | 377068 | 690 | 0.43 | 0.46 | 0.88 | CRU | 19990 | 117712 | 1144 | 0.5 | 0.43 | 0.95 |
| WOOL | 16874 | 351178 | 716 | 0.41 | 0.18 | 0.41 | POOH | 27245 | 111641 | 193 | 0.26 | 0.49 | 0.69 |
| DERC | 24277 | 111205 | 824 | 0.45 | 0.49 | 0.83 | aDAI | 13648 | 187050 | 1068 | 0.45 | 0.46 | 0.82 |
| stkAAVE | 37355 | 110924 | 1128 | 0.42 | 0.57 | 0.71 | ORN | 44010 | 239451 | 1134 | 0.46 | 0.47 | 0.87 |
| BTRFLY | 8450 | 108371 | 453 | 0.48 | 0.34 | 0.44 | DOGE2.0 | 7664 | 79047 | 123 | 0.45 | 0.38 | 0.66 |
| SDEX | 9127 | 104869 | 240 | 0.41 | 0.44 | 0.75 | HOICHI | 5075 | 77361 | 436 | 0.36 | 0.32 | 0.71 |
| XCN | 20085 | 104185 | 607 | 0.46 | 0.42 | 0.84 | EVERMOON | 7552 | 79868 | 163 | 0.24 | 0.35 | 0.52 |
| HOP | 37004 | 102650 | 514 | 0.41 | 0.6 | 0.88 | MUTE | 12426 | 82345 | 977 | 0.43 | 0.46 | 0.95 |
| MAHA | 18401 | 96180 | 749 | 0.43 | 0.47 | 0.91 | crvUSD | 2950 | 88647 | 174 | 0.61 | 0.37 | 0.73 |
| DINO | 15837 | 94140 | 358 | 0.44 | 0.44 | 0.74 | SLP | 6675 | 95368 | 1151 | 0.43 | 0.36 | 0.91 |
| bendWETH | 1454 | 96898 | 593 | 0.51 | 0.21 | 0.51 | sILV2 | 12838 | 92905 | 611 | 0.4 | 0.34 | 0.48 |
| PUSH | 14501 | 93103 | 936 | 0.46 | 0.38 | 0.83 | SPONGE | 25852 | 90468 | 184 | 0.31 | 0.66 | 0.81 |

We summarize detailed statistics of each token network in MiNT datasets in Table 5. Most networks have more than 10k nodes and over 100k edges. The lifespan of MiNT networks varies from 107 days to 6 years, and there exists at least one transaction each day. As the table shows, the token networks have quite high surprise values with an average of 0.82.

For our experiments, we follow the MiNT protocol and split the dataset into 64 financial networks for training and 20 additional networks for unseen testing. This setup allows Hydra to learn transferable temporal representations from diverse source networks and evaluate zero-shot generalization on new target networks that are only available at inference time.

## D BASELINES

In this section, we give further details about the temporal graph learning models we used as a baseline for our work.

**TGCN** (Zhao et al., 2020) is a combination of GCN and GRU. In particular, GCN is used to learn complex topological structures, while GRU is used to model embedding dynamically to capture temporal dependence.

**HTGN** (Yang et al., 2021) leverages the power of hyperbolic geometry, which is well-suited for capturing hierarchical structures and complex relationships in temporal networks. HTGN maps the temporal graph into hyperbolic space and utilizes hyperbolic graph neural networks and hyperbolic gated recurrent neural networks to model the evolving dynamics. It incorporates two key modules that are hyperbolic temporal contextual self-attention (HTA) and hyperbolic temporal consistency (HTC)-to ensure that temporal dependencies are effectively captured and that the model is both stable and generalizable across various tasks.

**GraphPulse** (Shamsi et al., 2024) addresses the challenge of learning from nodes and edges with different timestamps, which many existing models struggle with. It combines two key techniques: the Mapper method from topological data analysis to extract clustering information from graph nodes and Recurrent Neural Networks (RNNs) for temporal reasoning. This principled approach helps capture both the structure and dynamics of evolving graphs.

**GCLSTM** (Chen et al., 2022a) combines a Graph Convolutional Network (GCN) and Long Short-Term Memory (LSTM) units to handle both the structural and temporal aspects of evolving net-

works. The GCN is used to capture the local structural properties of the network at each snapshot, while the LSTM learns the temporal evolution of these snapshots over time.

**EvolveGCN** (Pareja et al., 2020) is designed to capture the temporal dynamics of graph-structured data. Instead of relying on static node embeddings, EvolveGCN evolves the parameters of a graph convolutional network (GCN) over time. By using a recurrent neural network (RNN) to adapt the GCN parameters, this model is capable of dynamically adjusting during both training and testing, allowing it to handle evolving graphs, even when node sets vary significantly across different time steps.

**ROLAND** (You et al., 2022) is a dynamic graph learning framework that models node representations as hierarchical states, updated recurrently to capture temporal dependencies in evolving graphs. It supports scalable training using techniques like truncated backpropagation through time and meta-learning. In our DTDG setting, we use ROLAND to benchmark its performance and adaptability across diverse temporal networks.

**WinGNN** (Zhu et al., 2023) uses a simple GNN to model topological information from the graph as other models existing in the literature. However, to model temporal dependencies, WinGNN proposes a novel mechanism of random gradient aggregation and meta learning strategy. In particular, WinGNN computes the frame-wise loss of the current snapshot and passes the loss gradient to the next to model graph dynamics without using RNN-based modules. Then it introduces the randomized sliding-window to acquire the windowaware gradienton consecutive snapshots, and the calculated two types of gradient are aggregated to update the GNN modules.

## E   TASK FORMALIZATIONS

In this section, we provide detailed definitions for all classification and regression tasks considered in `Hydra`. Each temporal snapshot corresponds to a 7-day interval, and the property prediction setup follows GraphPulse (Shamsi et al., 2024). We also note that some of these tasks and the corresponding labels were processed specifically for this work, ensuring consistency and comparability across networks.

Setting $n = 7$, $\delta_1 = 3$, and $\delta_2 = 10$ days, we establish a practical graph property with a 7-day prediction window. This choice is particularly relevant in financial contexts, such as Ethereum asset networks, where it can guide investment decisions (Abay et al., 2019).

### E.1   CLASSIFICATION TASKS

**Node Growth/Shrinkage (Node G/S).** Let $V(t_1, t_n)$ denote the set of unique nodes active between times $t_1$ and $t_n$. The task predicts whether the number of nodes increases in the prediction interval $[t_{n+\delta_1}, t_{n+\delta_2}]$:

$$P_{\text{nodes}}(\mathcal{G}, t_1, t_n, \delta_1, \delta_2) = \begin{cases} 1, & \text{if } |V(t_{n+\delta_1}, t_{n+\delta_2})| > |V(t_1, t_n)|, \\ 0, & \text{otherwise.} \end{cases} \tag{1}$$

**Importance.** Node growth measures adoption, reflecting the entry of new addresses, while shrinkage signals attrition. In token ecosystems, this corresponds to market expansion or decline.

**Edge Growth/Shrinkage (Edge G/S).** Let $E(t_1, t_n)$ denote the set of transactions in $[t_1, t_n]$. The model predicts whether transaction activity grows in the next interval:

$$P_{\text{edges}}(\mathcal{G}, t_1, t_n, \delta_1, \delta_2) = \begin{cases} 1, & \text{if } |E(t_{n+\delta_1}, t_{n+\delta_2})| > |E(t_1, t_n)|, \\ 0, & \text{otherwise.} \end{cases} \tag{2}$$

**Importance.** Edge growth captures changes in liquidity and market engagement, with direct implications for trading activity and token valuation.

**Largest Connected Component Growth/Shrinkage (LCC G/S).** Let $C(t_1, t_n)$ denote the size of the largest connected component during $[t_1, t_n]$. The task is to predict whether connectivity expands:

$$P_{\text{LCC}}(\mathcal{G}, t_1, t_n, \delta_1, \delta_2) = \begin{cases} 1, & \text{if } |C(t_{n+\delta_1}, t_{n+\delta_2})| > |C(t_1, t_n)|, \\ 0, & \text{otherwise.} \end{cases} \tag{3}$$

**Importance.** LCC growth reflects stronger structural integration, improving liquidity and market stability in blockchain networks.

## E.2 REGRESSION TASKS

**New Node Count.** Let $V_{\text{new}}(t_{n+\delta_1}, t_{n+\delta_2})$ denote the set of nodes that appear for the first time in the prediction interval. The regression target is:

$$\hat{y}_{\text{new-nodes}}(\mathcal{G}, t_1, t_n, \delta_1, \delta_2) = |V_{\text{new}}(t_{n+\delta_1}, t_{n+\delta_2})|. \tag{4}$$

**Importance.** New node prediction quantifies adoption and user acquisition, a key indicator of ecosystem growth.

**Edge Count.** The task is to predict the number of transactions in the future interval:

$$\hat{y}_{\text{edges}}(\mathcal{G}, t_1, t_n, \delta_1, \delta_2) = |E(t_{n+\delta_1}, t_{n+\delta_2})|. \tag{5}$$

**Importance.** Transaction forecasts provide fine-grained estimates of liquidity and demand surges in decentralized markets.

**Influential Node Count.** Define influential nodes as those with a degree greater than 5 in the prediction interval. Empirically, degree distributions of ERC-20 token graphs are heavy-tailed (Shamsi et al., 2025): most nodes appear only once or twice, and a small fraction appear hundreds or thousands of times. When we plot the cumulative distribution of node degrees, there is usually a sharp drop in frequency after degree 1–2, followed by a long but thinner tail. Setting the threshold at 5 sits just beyond this long tail cutoff, ensuring that only the top 10–20% of nodes in activity are retained as influential. In practice, this excludes wallets that perform only a handful of transfers while capturing the repeat participants who actually shape liquidity and flow.

Let $V_{\text{inf}}(t_{n+\delta_1}, t_{n+\delta_2}) = \{v \in V : \deg(v) > 5\}$. The regression target is:

$$\hat{y}_{\text{inf}}(\mathcal{G}, t_1, t_n, \delta_1, \delta_2) = |V_{\text{inf}}(t_{n+\delta_1}, t_{n+\delta_2})|. \tag{6}$$

**Importance.** Influential nodes represent hubs such as active traders, liquidity providers, or contracts, which shape the stability and price dynamics of token ecosystems.

## E.3 IMPORTANCE OF TEMPORAL PROPERTY PREDICTION

Monitoring global structural dynamics in temporal graphs is essential in domains where the evolution of the entire network, rather than individual edges, drives operational decisions. In blockchain transaction networks, most transactions occur only once and do not repeat, so link-level prediction is not informative. Graph properties instead track global indicators such as changes in connectivity, activity, and influential participants, which often serve as early signals of liquidity risk, instability, or ecosystem decline (Abay et al., 2019; Gurcan Akcora et al., 2021). Since thousands of Ethereum-based tokens evolve with their own user bases and activity patterns (Zhu et al., 2024), training a separate temporal model for each network is not practical. A single model that learns from many heterogeneous networks and jointly predicts several graph properties offers a more scalable and informative solution. Similar considerations arise in communication, payment, and social systems, where forecasting network-level behaviors helps detect anomalies, anticipate demand, and understand system health (Kazemi et al., 2020). A multi-task approach is particularly valuable because these properties move together and provide complementary signals, leading to more reliable and actionable predictions than solving each task independently.

# F EVALUATION METRICS

We evaluate models using both performance scores and ranking-based statistics. Below we provide formal definitions of each metric used in the main paper.

**First-Place Count.** For a given task, the *first-place count* measures how often a method achieves the top performance across datasets. Let $\mathcal{D}$ denote the set of datasets and $\mathcal{M}$ the set of methods. For dataset $d \in \mathcal{D}$, let $m^*(d) = \arg\max_{m \in \mathcal{M}} \text{Perf}(m, d)$, where $\text{Perf}(m, d)$ is the task-specific score (AUC for classification, MAE for regression). Then the first-place count for method $m$ is $\text{First}(m) = \sum_{d \in \mathcal{D}} \mathbf{1}[m = m^*(d)]$.

**Average Rank.** Each method is ranked per dataset according to performance. Let $\text{rank}(m, d)$ be the rank of method $m$ on dataset $d$ (lower is better). The average rank of method $m$ is $\text{AvgRank}(m) = \frac{1}{|\mathcal{D}|} \sum_{d \in \mathcal{D}} \text{rank}(m, d)$.

**Average AUC (Classification).** For binary classification, the Area Under the ROC Curve (AUC) for dataset $d$ is $\text{AUC}(d) = \Pr(\hat{y}_i > \hat{y}_j \mid y_i = 1, y_j = 0)$, where $\hat{y}_i$ are predicted scores and $y_i \in \{0, 1\}$ are ground-truth labels. The average AUC of method $m$ is $\text{AvgAUC}(m) = \frac{1}{|\mathcal{D}|} \sum_{d \in \mathcal{D}} \text{AUC}(m, d)$.

**Average MAE (Regression).** For regression, the Mean Absolute Error (MAE) on dataset $d$ is $\text{MAE}(d) = \frac{1}{N_d} \sum_{i=1}^{N_d} |\hat{y}_i - y_i|$, where $\hat{y}_i$ are predictions and $y_i$ ground truth labels. The average MAE of method $m$ is $\text{AvgMAE}(m) = \frac{1}{|\mathcal{D}|} \sum_{d \in \mathcal{D}} \text{MAE}(m, d)$.

**Relative Gain Across Tasks.** Relative gain quantifies Hydra's improvement over the strongest baseline by comparing the average metric values across all 20 test datasets for each task. For classification (where higher AUC is better), the gain for task $t$ is defined as

$$\text{Gain}_t = \frac{\overline{\text{AUC}}_{\text{Hydra}}(t) - \max_{m \in \mathcal{B}} \overline{\text{AUC}}_m(t)}{\max_{m \in \mathcal{B}} \overline{\text{AUC}}_m(t)} \times 100\%,$$

where $\overline{\text{AUC}}_m(t)$ denotes the average AUC of method $m$ on task $t$ across all test datasets.

For regression (where lower MAE is better), the gain is

$$\text{Gain}_t = \frac{\min_{m \in \mathcal{B}} \overline{\text{MAE}}_m(t) - \overline{\text{MAE}}_{\text{Hydra}}(t)}{\min_{m \in \mathcal{B}} \overline{\text{MAE}}_m(t)} \times 100\%,$$

where $\overline{\text{MAE}}_m(t)$ denotes the average MAE of method $m$ on task $t$ across all test datasets.

Finally, the overall gain across a group of tasks $\mathcal{T}$ (e.g., all classification tasks) is computed as the mean of the per-task gains: $\text{Gain}_{\mathcal{T}} = \frac{1}{|\mathcal{T}|} \sum_{t \in \mathcal{T}} \text{Gain}_t$.

## G ABLATION STUDIES

To better understand `Hydra` components, we conduct an ablation study on the edge growth/shrinkage task. The ablations reflect different stages in the model's incremental development. First, we compare the initial version of Hydra without pooling, evaluating the impact of adding Density of States (DOS) features in the spectral path. Next, we consider models that already include DOS and assess the effect of introducing attention-based pooling in the spatial path. Results are reported in Appendix Tables 6 and 7. Together, these comparisons show that DOS enhances the capture of global spectral structure, while pooling improves the selection of informative subgraphs, confirming their complementary roles in Hydra's design and performance.

**Impact of DOS.** To assess the contribution of the DOS module, we compare Hydra with and without DOS features (Table 6). This setup isolates the effect of spectral information on predictive performance, holding all other components fixed. The results show that incorporating DOS substantially improves performance on most datasets: Hydra with DOS achieves the best results in 14 out of 20 networks, compared to 6 without DOS. On average, DOS raises AUC from 0.697 to 0.719 and improves the mean rank from 1.70 to 1.30. Gains are particularly large on challenging datasets such as *EVERMOON*, *DOGE2.0*, and *DINO*, where DOS more effectively captures global spectral structure. These findings highlight that DOS provides complementary information to the spatial path, leading to stronger generalization across temporal networks.

**Impact of Attention-based Pooling.** The ablation on the edge growth/shrinkage task (Table 7) shows the contribution of the pooling mechanism within Hydra's spatial path. When pooling is enabled, the model achieves higher AUC on 17 out of 20 datasets, with the average score improving from 0.719 to 0.753 and the mean rank from 1.85 to 1.15. The improvement is particularly noticeable when pooling helps the spatial path focus on structurally important nodes, resulting in stronger graph-level representations. These findings indicate that pooling is an essential component for producing compact, informative graph-level representations and strengthen Hydra's ability to capture temporal dynamics.

Table 6: DOS ablation study on task Edge G/S. AUC values are reported. Best results per dataset are in **bold**.

| Dataset | Hydra w/o DOS | Hydra w DOS |
|---|---|---|
| MIR | **0.791 (±0.014)** | 0.755 (±0.013) |
| DOGE2.0 | 0.500 (±0.102) | **0.732 (±0.044)** |
| MUTE | 0.675 (±0.030) | **0.699 (±0.011)** |
| EVERMOON | 0.426 (±0.129) | **0.796 (±0.219)** |
| DERC | 0.748 (±0.029) | **0.761 (±0.030)** |
| ADX | 0.678 (±0.019) | **0.703 (±0.006)** |
| HOICHI | **0.792 (±0.052)** | 0.572 (±0.016) |
| SDEX | **0.599 (±0.134)** | 0.261 (±0.129) |
| BAG | 0.759 (±0.294) | **0.874 (±0.017)** |
| XCN | 0.822 (±0.073) | **0.825 (±0.038)** |
| ETH2x-FLI | 0.716 (±0.020) | **0.736 (±0.026)** |
| stkAAVE | 0.708 (±0.021) | **0.719 (±0.006)** |
| GLM | 0.755 (±0.194) | **0.810 (±0.015)** |
| QOM | 0.639 (±0.015) | **0.698 (±0.024)** |
| WOJAK | 0.548 (±0.081) | **0.633 (±0.081)** |
| DINO | 0.672 (±0.044) | **0.843 (±0.031)** |
| Metis | **0.785 (±0.050)** | 0.713 (±0.027) |
| REPv2 | **0.761 (±0.039)** | 0.736 (±0.028) |
| TRAC | **0.794 (±0.041)** | 0.732 (±0.007) |
| BEPRO | 0.775 (±0.022) | **0.786 (±0.009)** |
| 1st-Place Count ↑ | 6 | **14** |
| Avg. Rank ↓ | 1.70 | **1.30** |
| Avg. AUC ↑ | 0.697 | **0.719** |

Table 7: SAG pooling ablation study on task Edge G/S. AUC values are reported. Best results per dataset are in **bold**.

| Dataset | Hydra w/o Pooling | Hydra w Pooling |
|---|---|---|
| MIR | 0.755 (±0.013) | **0.793 (±0.002)** |
| DOGE2.0 | 0.732 (±0.044) | **0.897 (±0.089)** |
| MUTE | 0.699 (±0.011) | **0.701 (±0.098)** |
| EVERMOON | 0.796 (±0.219) | **0.818 (±0.046)** |
| DERC | 0.761 (±0.030) | **0.839 (±0.008)** |
| ADX | 0.703 (±0.006) | **0.722 (±0.087)** |
| HOICHI | 0.572 (±0.016) | **0.591 (±0.103)** |
| SDEX | 0.261 (±0.129) | **0.348 (±0.066)** |
| BAG | 0.874 (±0.017) | **0.969 (±0.008)** |
| XCN | 0.825 (±0.038) | **0.844 (±0.012)** |
| ETH2x-FLI | **0.736 (±0.026)** | 0.712 (±0.045) |
| stkAAVE | 0.719 (±0.006) | **0.732 (±0.011)** |
| GLM | 0.810 (±0.015) | **0.850 (±0.009)** |
| QOM | 0.698 (±0.024) | **0.745 (±0.003)** |
| WOJAK | **0.633 (±0.081)** | 0.585 (±0.067) |
| DINO | 0.843 (±0.031) | **0.895 (±0.003)** |
| Metis | 0.713 (±0.027) | **0.733 (±0.004)** |
| REPv2 | 0.736 (±0.028) | **0.772 (±0.016)** |
| TRAC | **0.732 (±0.007)** | 0.722 (±0.001) |
| BEPRO | 0.786 (±0.009) | **0.800 (±0.002)** |
| 1st-Place Count ↑ | 3 | **17** |
| Avg. Rank ↓ | 1.85 | **1.15** |
| Avg. AUC ↑ | 0.719 | **0.753** |

## H  SCALING BEHAVIOR IN HYDRA

We conducted scaling trend experiments to evaluate Hydra under different training pack sizes systematically. Four model variations were trained using 8, 16, and 32 networks, enabling us to observe how increasing the number of networks in the training loop affects performance. This setup provides a consistent method to study Hydra's behavior when exposed to varying amounts of training data across multiple tasks. For each task, we present results for all four Hydra variants. These results demonstrate how Hydra scales with the number of training networks and are shown in the following tables. Each table provides a detailed breakdown by task and model variation, offering a comprehensive view of performance under various scaling setups. The detailed results of each Hydra variation trained with different datapacks across all six tasks are reported in Table 8. Subsections (a)–(c) present classification tasks (Node G/S, Edge G/S, LCC G/S), while (d)–(f) correspond to regression tasks (New Node Count, Influential Node Count, Edge Count).

## I  EXTENDED NETWORK AND TASK RESULTS

This section reports the full per-network results for all classification and regression tasks, using the same evaluation protocol as in the main text. These tables complement the summary figures by showing the complete distribution of baseline and Hydra performance across datasets.

**Classification.** Extended classification results are provided for *Edge-G/S*, *LLC-G/S*, and *Node-G/S* in Table 9, Table 10, and Table 11, respectively.

**Regression.** Extended regression results are provided for *Edge Count*, *New Node Count*, and *Influential Node Count* in Table 12, Table 13, and Table 14, respectively.

For classification, Hydra achieves the highest AUC on the majority of datasets across Edge-G/S, LLC-G/S, and Node-G/S. For regression Hydra consistently delivers the lowest or near-lowest MAE, remaining competitive even on tasks where another baseline occasionally leads. These results confirm that Hydra's advantages are not limited to averages: the model transfers robustly across heterogeneous temporal networks and maintains strong performance in the zero-shot setting.

Table 8: Performance of `Hydra` across six tasks as the number of training networks increases from 8 to 32.

(a) Classification : Node G/S

| Dataset | Hydra-8 | Hydra-16 | Hydra-32 |
|---|---|---|---|
| MIR | **0.769 (±0.003)** | **0.769 (±0.007)** | 0.764 (±0.001) |
| DOGE2.0 | 0.613 (±0.083) | 0.573 (±0.133) | **0.633 (±0.064)** |
| MUTE | 0.755 (±0.016) | **0.763 (±0.003)** | 0.748 (±0.025) |
| EVERMOON | 0.624 (±0.005) | 0.582 (±0.031) | **0.655 (±0.040)** |
| DERC | 0.734 (±0.010) | 0.725 (±0.006) | **0.742 (±0.004)** |
| ADX | 0.753 (±0.027) | **0.767 (±0.004)** | 0.718 (±0.051) |
| HOICHI | **0.582 (±0.029)** | 0.552 (±0.050) | 0.558 (±0.047) |
| SDEX | **0.762 (±0.008)** | 0.748 (±0.068) | 0.743 (±0.026) |
| BAG | 0.963 (±0.010) | 0.952 (±0.021) | **0.969 (±0.009)** |
| XCN | 0.862 (±0.017) | 0.821 (±0.070) | **0.878 (±0.009)** |
| ETH2x-FLI | **0.717 (±0.030)** | 0.710 (±0.009) | 0.678 (±0.031) |
| stkAAVE | 0.776 (±0.009) | 0.776 (±0.003) | **0.779 (±0.007)** |
| GLM | 0.750 (±0.011) | 0.746 (±0.010) | **0.763 (±0.005)** |
| QOM | 0.703 (±0.012) | 0.707 (±0.013) | **0.719 (±0.012)** |
| WOJAK | **0.502 (±0.103)** | 0.352 (±0.157) | 0.412 (±0.060) |
| DINO | 0.903 (±0.029) | 0.905 (±0.021) | **0.910 (±0.013)** |
| Metis | 0.685 (±0.002) | 0.632 (±0.091) | **0.693 (±0.007)** |
| REPv2 | **0.728 (±0.022)** | 0.721 (±0.023) | 0.689 (±0.013) |
| TRAC | 0.756 (±0.005) | 0.745 (±0.014) | **0.765 (±0.003)** |
| BEPRO | 0.858 (±0.023) | 0.806 (±0.094) | **0.865 (±0.010)** |
| **1st-Place Count↑** | 6 | 3 | **12** |
| **Avg. Rank↓** | 1.85 | 2.45 | **1.70** |
| **Avg. AUC↑** | **0.740** | 0.718 | 0.734 |

(b) Classification : Edge G/S

| Dataset | Hydra-8 | Hydra-16 | Hydra-32 |
|---|---|---|---|
| MIR | **0.800 (±0.007)** | 0.796 (±0.001) | 0.793 (±0.002) |
| DOGE2.0 | 0.859 (±0.022) | **0.897 (±0.089)** | **0.897 (±0.089)** |
| MUTE | 0.757 (±0.004) | **0.764 (±0.004)** | 0.701 (±0.098) |
| EVERMOON | 0.698 (±0.180) | 0.750 (±0.024) | **0.818 (±0.046)** |
| DERC | 0.823 (±0.018) | 0.826 (±0.002) | **0.839 (±0.008)** |
| ADX | 0.760 (±0.004) | **0.768 (±0.015)** | 0.722 (±0.087) |
| HOICHI | 0.606 (±0.015) | **0.617 (±0.026)** | 0.591 (±0.103) |
| SDEX | 0.288 (±0.023) | 0.331 (±0.049) | **0.348 (±0.066)** |
| BAG | **0.969 (±0.009)** | 0.955 (±0.015) | **0.969 (±0.008)** |
| XCN | **0.847 (±0.006)** | 0.845 (±0.010) | 0.844 (±0.012) |
| ETH2x-FLI | 0.735 (±0.006) | **0.738 (±0.005)** | 0.712 (±0.045) |
| stkAAVE | **0.744 (±0.008)** | 0.743 (±0.007) | 0.732 (±0.011) |
| GLM | 0.849 (±0.003) | 0.841 (±0.015) | **0.850 (±0.009)** |
| QOM | 0.755 (±0.015) | **0.762 (±0.024)** | 0.745 (±0.003) |
| WOJAK | **0.627 (±0.014)** | 0.561 (±0.101) | 0.585 (±0.067) |
| DINO | 0.875 (±0.030) | 0.889 (±0.009) | **0.895 (±0.003)** |
| Metis | **0.735 (±0.004)** | 0.697 (±0.062) | 0.733 (±0.004) |
| REPv2 | 0.778 (±0.001) | **0.784 (±0.009)** | 0.772 (±0.016) |
| TRAC | 0.713 (±0.010) | 0.711 (±0.011) | **0.722 (±0.001)** |
| BEPRO | 0.783 (±0.009) | 0.743 (±0.085) | **0.800 (±0.002)** |
| **1st-Place Count↑** | 6 | 7 | **9** |
| **Avg. Rank↓** | **1.95** | **1.95** | 2 |
| **Avg. AUC↑** | 0.75 | 0.751 | **0.753** |

(c) Classification : LCC G/S

| Dataset | Hydra-8 | Hydra-16 | Hydra-32 |
|---|---|---|---|
| MIR | **0.819 (±0.004)** | 0.817 (±0.001) | 0.815 (±0.007) |
| DOGE2.0 | **0.762 (±0.010)** | 0.667 (±0.175) | 0.702 (±0.160) |
| MUTE | 0.714 (±0.049) | **0.756 (±0.006)** | 0.704 (±0.085) |
| EVERMOON | 0.600 (±0.057) | 0.639 (±0.029) | **0.667 (±0.045)** |
| DERC | **0.819 (±0.003)** | 0.813 (±0.005) | 0.808 (±0.022) |
| ADX | 0.691 (±0.066) | **0.770 (±0.021)** | 0.727 (±0.093) |
| HOICHI | 0.610 (±0.044) | **0.640 (±0.041)** | 0.638 (±0.075) |
| SDEX | 0.784 (±0.017) | 0.810 (±0.048) | **0.816 (±0.025)** |
| BAG | 0.968 (±0.013) | 0.957 (±0.021) | **0.976 (±0.006)** |
| XCN | 0.863 (±0.011) | 0.863 (±0.029) | **0.887 (±0.014)** |
| ETH2x-FLI | **0.708 (±0.011)** | 0.703 (±0.001) | 0.687 (±0.032) |
| stkAAVE | 0.753 (±0.011) | **0.757 (±0.003)** | 0.748 (±0.012) |
| GLM | **0.853 (±0.006)** | 0.844 (±0.020) | 0.848 (±0.012) |
| QOM | 0.725 (±0.009) | **0.730 (±0.002)** | 0.729 (±0.010) |
| WOJAK | **0.558 (±0.032)** | 0.458 (±0.142) | 0.500 (±0.083) |
| DINO | 0.878 (±0.021) | **0.883 (±0.032)** | 0.818 (±0.049) |
| Metis | 0.727 (±0.006) | 0.669 (±0.100) | **0.731 (±0.001)** |
| REPv2 | 0.781 (±0.004) | **0.785 (±0.009)** | 0.764 (±0.020) |
| TRAC | 0.772 (±0.006) | 0.768 (±0.008) | **0.781 (±0.004)** |
| BEPRO | 0.826 (±0.015) | 0.800 (±0.058) | **0.830 (±0.006)** |
| **1st-Place Count↑** | 6 | **7** | **7** |
| **Avg. Rank↓** | **1.95** | 2 | 2 |
| **Avg. AUC↑** | **0.761** | 0.756 | 0.759 |

(d) Regression : New Node Count

| Dataset | Hydra-8 | Hydra-16 | Hydra-32 |
|---|---|---|---|
| MIR | 0.056 (±0.057) | 0.040 (±0.042) | **0.013 (±0.004)** |
| DOGE2.0 | 0.068 (±0.023) | **0.060 (±0.027)** | 0.092 (±0.008) |
| MUTE | 0.073 (±0.036) | 0.036 (±0.025) | **0.025 (±0.005)** |
| EVERMOON | 0.070 (±0.053) | 0.038 (±0.045) | **0.012 (±0.005)** |
| DERC | 0.071 (±0.049) | 0.038 (±0.040) | **0.015 (±0.004)** |
| ADX | 0.069 (±0.047) | 0.029 (±0.035) | **0.016 (±0.005)** |
| HOICHI | 0.064 (±0.038) | 0.048 (±0.030) | **0.029 (±0.008)** |
| SDEX | 0.074 (±0.021) | 0.072 (±0.011) | **0.063 (±0.006)** |
| BAG | 0.072 (±0.023) | 0.056 (±0.009) | **0.052 (±0.002)** |
| XCN | 0.067 (±0.056) | 0.036 (±0.044) | **0.009 (±0.002)** |
| ETH2x-FLI | 0.053 (±0.034) | 0.032 (±0.023) | **0.030 (±0.004)** |
| stkAAVE | 0.085 (±0.037) | **0.057 (±0.024)** | 0.078 (±0.017) |
| GLM | 0.097 (±0.013) | 0.103 (±0.008) | **0.094 (±0.007)** |
| QOM | 0.073 (±0.054) | 0.034 (±0.040) | **0.012 (±0.001)** |
| WOJAK | 0.076 (±0.059) | 0.034 (±0.042) | **0.009 (±0.000)** |
| DINO | 0.065 (±0.053) | 0.046 (±0.042) | **0.018 (±0.007)** |
| Metis | 0.072 (±0.030) | 0.037 (±0.016) | **0.034 (±0.010)** |
| REPv2 | 0.083 (±0.024) | 0.074 (±0.015) | **0.066 (±0.004)** |
| TRAC | 0.074 (±0.042) | 0.029 (±0.026) | **0.022 (±0.005)** |
| BEPRO | 0.081 (±0.067) | 0.034 (±0.046) | **0.004 (±0.001)** |
| **1st-Place Count↑** | 0 | 2 | **18** |
| **Avg. Rank↓** | 2.90 | 1.95 | **1.15** |
| **Avg. MAE↓** | 0.072 | 0.047 | **0.035** |

(e) Regression : Influential Node Count

| Dataset | Hydra-8 | Hydra-16 | Hydra-32 |
|---|---|---|---|
| MIR | 0.080 (±0.059) | 0.059 (±0.004) | **0.039 (±0.005)** |
| DOGE2.0 | **0.065 (±0.034)** | 0.094 (±0.003) | 0.126 (±0.032) |
| MUTE | 0.103 (±0.070) | 0.078 (±0.034) | **0.045 (±0.027)** |
| EVERMOON | 0.107 (±0.067) | 0.072 (±0.018) | **0.038 (±0.024)** |
| DERC | 0.063 (±0.065) | 0.056 (±0.012) | **0.033 (±0.017)** |
| ADX | 0.084 (±0.070) | 0.054 (±0.009) | **0.030 (±0.001)** |
| HOICHI | 0.115 (±0.066) | 0.088 (±0.045) | **0.055 (±0.028)** |
| SDEX | 0.087 (±0.027) | 0.083 (±0.068) | **0.042 (±0.019)** |
| BAG | 0.141 (±0.064) | 0.109 (±0.045) | **0.075 (±0.027)** |
| XCN | **0.071 (±0.033)** | 0.096 (±0.008) | 0.126 (±0.033) |
| ETH2x-FLI | 0.066 (±0.059) | 0.052 (±0.007) | **0.033 (±0.020)** |
| stkAAVE | 0.067 (±0.049) | 0.063 (±0.008) | **0.042 (±0.013)** |
| GLM | 0.108 (±0.045) | 0.088 (±0.014) | **0.075 (±0.008)** |
| QOM | 0.066 (±0.055) | 0.055 (±0.004) | **0.039 (±0.020)** |
| WOJAK | 0.116 (±0.068) | 0.074 (±0.006) | **0.036 (±0.027)** |
| DINO | 0.085 (±0.066) | 0.059 (±0.009) | **0.034 (±0.002)** |
| Metis | 0.082 (±0.056) | 0.072 (±0.022) | **0.046 (±0.008)** |
| REPv2 | **0.127 (±0.015)** | 0.130 (±0.004) | 0.129 (±0.022) |
| TRAC | 0.069 (±0.053) | 0.059 (±0.003) | **0.043 (±0.023)** |
| BEPRO | 0.105 (±0.068) | 0.064 (±0.005) | **0.033 (±0.018)** |
| **1st-Place Count↑** | 3 | 0 | **17** |
| **Avg. Rank↓** | 2.70 | 2.05 | **1.25** |
| **Avg. MAE↓** | 0.090 | 0.075 | **0.056** |

(f) Regression : Edge Count

| Dataset | Hydra-8 | Hydra-16 | Hydra-32 |
|---|---|---|---|
| MIR | 0.087 (±0.057) | 0.035 (±0.027) | **0.016 (±0.004)** |
| DOGE2.0 | **0.115 (±0.023)** | 0.135 (±0.032) | 0.187 (±0.008) |
| MUTE | 0.092 (±0.036) | 0.048 (±0.017) | **0.021 (±0.005)** |
| EVERMOON | 0.085 (±0.053) | 0.041 (±0.029) | **0.017 (±0.005)** |
| DERC | 0.082 (±0.049) | 0.033 (±0.024) | **0.021 (±0.004)** |
| ADX | 0.082 (±0.047) | 0.029 (±0.018) | **0.025 (±0.005)** |
| HOICHI | 0.090 (±0.038) | 0.062 (±0.006) | **0.027 (±0.008)** |
| SDEX | 0.080 (±0.021) | **0.077 (±0.018)** | 0.095 (±0.006) |
| BAG | 0.099 (±0.023) | 0.065 (±0.005) | **0.041 (±0.002)** |
| XCN | 0.066 (±0.056) | **0.027 (±0.009)** | 0.062 (±0.002) |
| ETH2x-FLI | 0.084 (±0.034) | 0.035 (±0.019) | **0.023 (±0.004)** |
| stkAAVE | 0.065 (±0.037) | **0.025 (±0.004)** | 0.051 (±0.017) |
| GLM | 0.106 (±0.013) | 0.101 (±0.007) | **0.087 (±0.007)** |
| QOM | 0.074 (±0.054) | **0.032 (±0.019)** | 0.036 (±0.001) |
| WOJAK | 0.090 (±0.059) | 0.034 (±0.035) | **0.010 (±0.000)** |
| DINO | 0.085 (±0.053) | 0.040 (±0.028) | **0.017 (±0.007)** |
| Metis | 0.087 (±0.030) | 0.041 (±0.005) | **0.034 (±0.010)** |
| REPv2 | 0.127 (±0.024) | **0.105 (±0.003)** | 0.111 (±0.004) |
| TRAC | 0.086 (±0.042) | 0.029 (±0.020) | **0.021 (±0.005)** |
| BEPRO | 0.094 (±0.067) | 0.037 (±0.036) | **0.011 (±0.001)** |
| **1st-Place Count↑** | 1 | 5 | **14** |
| **Avg. Rank↓** | 2.85 | 1.75 | **1.40** |
| **Avg. MAE↓** | 0.089 | 0.052 | **0.046** |

Table 9: AUC results for the Edge Growth/Shrinkage prediction task ( classification ). Best results are in **bold**, second best are underlined.

| Dataset | Single Model on Individual Networks | | | | | | | Transfer Models | |
|---|---|---|---|---|---|---|---|---|---|
| | HTGN | GC-LSTM | EvolveGCN | GraphPulse | ROLAND | TGCN | WinGNN | MiNT | Hydra (Ours) |
| MIR | 0.750 ±0.005 | 0.768 ±0.026 | 0.745 ±0.015 | 0.689 ±0.097 | 0.228 ±0.060 | 0.749 ±0.026 | 0.742 ±0.015 | **0.836** ±0.016 | 0.793 ±0.026 |
| DOGE2.0 | 0.590 ±0.059 | 0.538 ±0.000 | 0.551 ±0.022 | 0.384 ±0.180 | 0.513 ±0.022 | 0.487 ±0.044 | 0.577 ±0.038 | 0.538 ±0.038 | **0.897** ±0.044 |
| MUTE | 0.649 ±0.015 | 0.593 ±0.030 | 0.617 ±0.010 | **0.779** ±0.004 | 0.289 ±0.042 | 0.557 ±0.068 | 0.593 ±0.054 | 0.673 ±0.013 | 0.701 ±0.068 |
| EVERMOON | 0.512 ±0.023 | 0.562 ±0.179 | 0.451 ±0.046 | 0.519 ±0.130 | 0.349 ±0.119 | 0.463 ±0.149 | 0.525 ±0.114 | 0.517 ±0.039 | **0.818** ±0.149 |
| DERC | 0.683 ±0.013 | 0.703 ±0.022 | 0.669 ±0.009 | 0.769 ±0.040 | 0.405 ±0.357 | 0.743 ±0.077 | 0.674 ±0.044 | 0.798 ±0.027 | **0.839** ±0.077 |
| ADX | 0.769 ±0.018 | 0.723 ±0.002 | 0.718 ±0.004 | **0.784** ±0.002 | 0.761 ±0.011 | 0.674 ±0.034 | 0.733 ±0.023 | 0.679 ±0.024 | 0.722 ±0.034 |
| HOICHI | 0.807 ±0.047 | **0.857** ±0.000 | 0.856 ±0.001 | 0.714 ±0.010 | 0.815 ±0.036 | 0.836 ±0.034 | 0.769 ±0.101 | 0.765 ±0.018 | 0.591 ±0.034 |
| SDEX | **0.762** ±0.034 | 0.720 ±0.002 | 0.733 ±0.028 | 0.436 ±0.030 | 0.483 ±0.254 | 0.759 ±0.039 | 0.726 ±0.000 | 0.614 ±0.020 | 0.348 ±0.039 |
| BAG | 0.673 ±0.227 | 0.196 ±0.179 | 0.329 ±0.040 | 0.934 ±0.020 | 0.418 ±0.016 | 0.334 ±0.171 | 0.485 ±0.105 | 0.931 ±0.028 | **0.969** ±0.171 |
| XCN | 0.668 ±0.099 | 0.306 ±0.092 | 0.512 ±0.067 | 0.821 ±0.004 | 0.765 ±0.015 | 0.703 ±0.037 | 0.586 ±0.029 | **0.851** ±0.043 | 0.844 ±0.037 |
| ETH2x-FLI | 0.610 ±0.059 | 0.670 ±0.009 | 0.688 ±0.010 | 0.666 ±0.047 | 0.621 ±0.023 | 0.647 ±0.020 | 0.617 ±0.056 | **0.729** ±0.015 | 0.712 ±0.020 |
| stkAAVE | 0.702 ±0.042 | 0.368 ±0.011 | 0.397 ±0.022 | **0.743** ±0.006 | 0.591 ±0.122 | 0.577 ±0.129 | 0.572 ±0.018 | 0.709 ±0.022 | 0.732 ±0.129 |
| GLM | 0.830 ±0.029 | 0.451 ±0.003 | 0.501 ±0.033 | 0.769 ±0.018 | 0.559 ±0.357 | 0.531 ±0.008 | 0.530 ±0.004 | 0.831 ±0.024 | **0.850** ±0.008 |
| QOM | 0.633 ±0.017 | 0.612 ±0.001 | 0.618 ±0.002 | **0.775** ±0.011 | 0.641 ±0.003 | 0.647 ±0.032 | 0.645 ±0.099 | 0.647 ±0.019 | 0.745 ±0.032 |
| WOJAK | 0.479 ±0.005 | 0.484 ±0.000 | 0.505 ±0.023 | 0.467 ±0.030 | 0.529 ±0.005 | 0.516 ±0.021 | 0.511 ±0.026 | 0.524 ±0.027 | **0.585** ±0.021 |
| DINO | 0.730 ±0.195 | 0.874 ±0.028 | 0.868 ±0.029 | 0.801 ±0.020 | 0.497 ±0.092 | 0.544 ±0.314 | 0.628 ±0.251 | 0.779 ±0.113 | **0.895** ±0.314 |
| Metis | 0.715 ±0.122 | 0.646 ±0.023 | 0.688 ±0.027 | **0.812** ±0.011 | 0.696 ±0.108 | 0.709 ±0.033 | 0.690 ±0.039 | 0.760 ±0.025 | 0.733 ±0.033 |
| REPv2 | 0.760 ±0.012 | 0.725 ±0.014 | 0.709 ±0.002 | **0.830** ±0.001 | 0.751 ±0.003 | 0.696 ±0.035 | 0.744 ±0.026 | 0.789 ±0.020 | 0.772 ±0.035 |
| TRAC | 0.712 ±0.071 | 0.748 ±0.000 | 0.748 ±0.026 | 0.767 ±0.001 | 0.495 ±0.223 | 0.741 ±0.012 | 0.752 ±0.007 | **0.785** ±0.008 | 0.722 ±0.012 |
| BEPRO | 0.655 ±0.038 | 0.632 ±0.019 | 0.610 ±0.012 | 0.783 ±0.003 | 0.439 ±0.125 | 0.744 ±0.074 | 0.736 ±0.018 | 0.782 ±0.003 | **0.800** ±0.074 |
| 1st-Place Count↑ | 1 | 1 | 0 | 6 | 0 | 0 | 0 | 4 | **8** |
| Avg. Rank ↓ | 4.85 | 5.80 | 6.10 | 3.80 | 6.30 | 5.60 | 5.55 | 3.30 | **2.80** |
| Avg. AUC ↑ | 0.684 | 0.609 | 0.626 | 0.712 | 0.542 | 0.633 | 0.642 | 0.727 | **0.753** |

Table 10: AUC results for the LCC Growth/Shrinkage prediction task ( classification ). Best results are in **bold**, second best are underlined.

| Dataset | Single Model on Individual Networks | | | | | | | Transfer Models | |
|---|---|---|---|---|---|---|---|---|---|
| | HTGN | GC-LSTM | EvolveGCN | GraphPulse | ROLAND | TGCN | WinGNN | MiNT | Hydra (Ours) |
| MIR | 0.745 (±0.023) | 0.585 (±0.128) | 0.575 (±0.146) | 0.800 (±0.008) | 0.536 (±0.275) | 0.585 (±0.055) | 0.749 (±0.020) | **0.845** (±0.035) | 0.815 (±0.007) |
| DOGE2.0 | 0.446 (±0.164) | 0.387 (±0.294) | 0.583 (±0.115) | 0.333 (±0.042) | 0.411 (±0.232) | 0.464 (±0.182) | 0.595 (±0.176) | 0.661 (±0.047) | **0.702** (±0.160) |
| MUTE | 0.574 (±0.022) | 0.579 (±0.022) | 0.578 (±0.033) | 0.647 (±0.014) | 0.624 (±0.037) | 0.567 (±0.007) | 0.641 (±0.061) | 0.582 (±0.078) | **0.704** (±0.085) |
| EVERMOON | 0.494 (±0.127) | 0.512 (±0.112) | 0.548 (±0.152) | 0.463 (±0.034) | 0.491 (±0.157) | 0.624 (±0.004) | 0.603 (±0.041) | 0.527 (±0.118) | **0.667** (±0.045) |
| DERC | 0.717 (±0.035) | 0.591 (±0.010) | 0.553 (±0.044) | 0.727 (±0.009) | 0.481 (±0.131) | 0.523 (±0.103) | 0.582 (±0.043) | 0.689 (±0.096) | **0.808** (±0.022) |
| ADX | **0.753** (±0.013) | 0.599 (±0.012) | 0.604 (±0.030) | 0.661 (±0.006) | 0.606 (±0.059) | 0.621 (±0.017) | 0.611 (±0.062) | 0.587 (±0.014) | 0.727 (±0.093) |
| HOICHI | 0.746 (±0.010) | 0.749 (±0.001) | 0.745 (±0.003) | 0.730 (±0.017) | 0.360 (±0.121) | **0.750** (±0.002) | 0.635 (±0.183) | 0.722 (±0.034) | 0.638 (±0.075) |
| SDEX | **0.911** (±0.104) | 0.721 (±0.138) | 0.601 (±0.105) | 0.808 (±0.050) | 0.825 (±0.047) | 0.770 (±0.231) | 0.575 (±0.282) | 0.382 (±0.280) | 0.816 (±0.025) |
| BAG | 0.493 (±0.043) | 0.291 (±0.180) | 0.480 (±0.052) | 0.900 (±0.010) | 0.463 (±0.019) | 0.463 (±0.141) | 0.490 (±0.080) | 0.893 (±0.074) | **0.976** (±0.006) |
| XCN | 0.566 (±0.199) | 0.481 (±0.160) | 0.533 (±0.257) | 0.681 (±0.005) | 0.569 (±0.204) | 0.638 (±0.045) | 0.549 (±0.133) | 0.827 (±0.025) | **0.887** (±0.014) |
| ETH2x-FLI | 0.561 (±0.037) | 0.529 (±0.017) | 0.547 (±0.009) | 0.653 (±0.047) | 0.499 (±0.135) | 0.549 (±0.019) | 0.505 (±0.090) | 0.618 (±0.025) | **0.687** (±0.032) |
| stkAAVE | 0.623 (±0.077) | 0.581 (±0.085) | 0.551 (±0.102) | 0.662 (±0.004) | 0.532 (±0.140) | 0.543 (±0.102) | 0.489 (±0.105) | 0.688 (±0.019) | **0.748** (±0.012) |
| GLM | 0.761 (±0.031) | 0.481 (±0.073) | 0.636 (±0.123) | 0.749 (±0.014) | 0.802 (±0.037) | 0.425 (±0.005) | 0.489 (±0.079) | 0.818 (±0.074) | **0.848** (±0.012) |
| QOM | 0.658 (±0.150) | 0.509 (±0.100) | 0.562 (±0.022) | **0.747** (±0.006) | 0.627 (±0.134) | 0.419 (±0.044) | 0.546 (±0.152) | 0.645 (±0.109) | 0.729 (±0.010) |
| WOJAK | 0.378 (±0.028) | 0.489 (±0.133) | 0.394 (±0.079) | **0.550** (±0.036) | 0.360 (±0.005) | 0.481 (±0.092) | 0.415 (±0.017) | 0.492 (±0.107) | 0.500 (±0.083) |
| DINO | 0.706 (±0.120) | 0.796 (±0.023) | 0.710 (±0.034) | 0.661 (±0.026) | 0.523 (±0.238) | 0.773 (±0.043) | 0.731 (±0.037) | 0.561 (±0.040) | **0.818** (±0.049) |
| Metis | 0.679 (±0.039) | 0.687 (±0.018) | 0.672 (±0.016) | **0.783** (±0.007) | 0.672 (±0.103) | 0.657 (±0.014) | 0.634 (±0.042) | 0.780 (±0.041) | 0.731 (±0.001) |
| REPv2 | 0.730 (±0.007) | 0.653 (±0.015) | 0.644 (±0.027) | 0.752 (±0.001) | 0.658 (±0.103) | 0.646 (±0.025) | 0.683 (±0.014) | 0.742 (±0.041) | **0.764** (±0.020) |
| TRAC | 0.733 (±0.043) | 0.629 (±0.017) | 0.623 (±0.004) | 0.686 (±0.001) | 0.606 (±0.117) | 0.620 (±0.005) | 0.599 (±0.026) | 0.762 (±0.028) | **0.781** (±0.004) |
| BEPRO | 0.694 (±0.009) | 0.595 (±0.008) | 0.557 (±0.058) | 0.725 (±0.004) | 0.482 (±0.146) | 0.536 (±0.031) | 0.582 (±0.063) | 0.628 (±0.017) | **0.830** (±0.006) |
| 1st-Place Count↑ | 2 | 0 | 0 | 3 | 0 | 1 | 0 | 1 | **13** |
| Avg. Rank ↓ | 4.40 | 6.00 | 6.30 | 3.50 | 6.95 | 6.10 | 6.10 | 3.95 | **1.70** |
| Avg. AUC ↑ | 0.648 | 0.572 | 0.585 | 0.686 | 0.556 | 0.583 | 0.585 | 0.672 | **0.759** |

Table 11: AUC results for the Node Growth/Shrinkage prediction task ( classification ). Best results are in **bold**, second best are underlined.

| | Single Model on Individual Networks | | | | | | | Transfer Models | |
|---|---|---|---|---|---|---|---|---|---|
| Dataset | HTGN | GC-LSTM | EvolveGCN | GraphPulse | ROLAND | TGCN | WinGNN | MiNT | Hydra (Ours) |
| MIR | 0.545 (±0.030) | 0.537 (±0.033) | 0.528 (±0.081) | 0.633 (±0.066) | 0.472 (±0.040) | 0.532 (±0.021) | 0.525 (±0.043) | 0.622 (±0.030) | **0.764** (±0.001) |
| DOGE2.0 | 0.427 (±0.065) | 0.633 (±0.014) | 0.400 (±0.061) | 0.403 (±0.035) | 0.260 (±0.000) | 0.627 (±0.034) | 0.693 (±0.041) | **0.750** (±0.014) | 0.633 (±0.064) |
| MUTE | 0.518 (±0.023) | 0.448 (±0.008) | 0.475 (±0.051) | 0.677 (±0.008) | 0.411 (±0.053) | 0.458 (±0.009) | 0.562 (±0.015) | 0.606 (±0.056) | **0.748** (±0.025) |
| EVERMOON | 0.585 (±0.059) | 0.606 (±0.021) | 0.488 (±0.088) | 0.463 (±0.002) | 0.427 (±0.137) | 0.567 (±0.030) | 0.548 (±0.116) | 0.614 (±0.071) | **0.655** (±0.040) |
| DERC | 0.662 (±0.051) | 0.492 (±0.069) | 0.503 (±0.084) | 0.611 (±0.049) | 0.551 (±0.013) | 0.447 (±0.003) | 0.517 (±0.034) | 0.569 (±0.004) | **0.742** (±0.004) |
| ADX | 0.678 (±0.017) | 0.505 (±0.043) | 0.509 (±0.022) | 0.701 (±0.003) | 0.557 (±0.082) | 0.484 (±0.048) | 0.504 (±0.018) | 0.507 (±0.037) | **0.718** (±0.051) |
| HOICHI | 0.687 (±0.004) | 0.718 (±0.007) | 0.685 (±0.020) | **0.745** (±0.006) | 0.347 (±0.084) | 0.718 (±0.002) | 0.526 (±0.188) | 0.492 (±0.120) | 0.558 (±0.047) |
| SDEX | 0.824 (±0.106) | 0.364 (±0.148) | 0.817 (±0.032) | **0.865** (±0.011) | 0.779 (±0.018) | 0.755 (±0.202) | 0.757 (±0.072) | 0.861 (±0.025) | 0.743 (±0.026) |
| BAG | 0.735 (±0.075) | 0.337 (±0.089) | 0.166 (±0.066) | 0.897 (±0.016) | 0.390 (±0.088) | 0.391 (±0.219) | 0.515 (±0.008) | 0.685 (±0.038) | **0.969** (±0.009) |
| XCN | 0.476 (±0.012) | 0.466 (±0.012) | 0.407 (±0.176) | 0.671 (±0.020) | 0.430 (±0.144) | 0.483 (±0.036) | 0.355 (±0.017) | 0.505 (±0.002) | **0.878** (±0.009) |
| ETH2x-FLI | 0.628 (±0.022) | 0.548 (±0.002) | 0.548 (±0.002) | 0.615 (±0.020) | 0.488 (±0.063) | 0.553 (±0.036) | 0.586 (±0.098) | 0.411 (±0.066) | **0.678** (±0.031) |
| stkAAVE | 0.517 (±0.093) | 0.543 (±0.043) | 0.456 (±0.069) | 0.643 (±0.005) | 0.661 (±0.037) | 0.425 (±0.029) | 0.465 (±0.036) | 0.561 (±0.007) | **0.779** (±0.007) |
| GLM | 0.706 (±0.014) | 0.566 (±0.001) | 0.516 (±0.105) | 0.595 (±0.003) | 0.493 (±0.149) | 0.575 (±0.019) | 0.610 (±0.024) | 0.720 (±0.045) | **0.763** (±0.005) |
| QOM | 0.647 (±0.094) | 0.492 (±0.003) | 0.485 (±0.001) | 0.705 (±0.002) | 0.592 (±0.080) | 0.495 (±0.004) | 0.409 (±0.051) | 0.572 (±0.017) | **0.719** (±0.012) |
| WOJAK | 0.417 (±0.143) | 0.338 (±0.068) | 0.357 (±0.104) | 0.500 (±0.000) | 0.202 (±0.018) | 0.488 (±0.080) | 0.314 (±0.029) | **0.618** (±0.035) | 0.412 (±0.060) |
| DINO | 0.845 (±0.015) | 0.323 (±0.148) | 0.444 (±0.052) | 0.686 (±0.007) | 0.330 (±0.115) | 0.615 (±0.070) | 0.600 (±0.292) | 0.735 (±0.005) | **0.910** (±0.013) |
| Metis | 0.589 (±0.049) | 0.483 (±0.052) | 0.566 (±0.012) | 0.652 (±0.029) | 0.574 (±0.040) | 0.549 (±0.011) | 0.510 (±0.005) | 0.616 (±0.012) | **0.693** (±0.007) |
| REPv2 | 0.650 (±0.004) | 0.519 (±0.023) | 0.515 (±0.019) | 0.662 (±0.008) | 0.597 (±0.028) | 0.534 (±0.029) | 0.626 (±0.058) | **0.710** (±0.129) | 0.689 (±0.013) |
| TRAC | 0.670 (±0.031) | 0.527 (±0.016) | 0.524 (±0.003) | 0.610 (±0.055) | 0.546 (±0.024) | 0.524 (±0.001) | 0.528 (±0.020) | 0.600 (±0.027) | **0.765** (±0.003) |
| BEPRO | 0.500 (±0.055) | 0.332 (±0.010) | 0.356 (±0.015) | 0.707 (±0.005) | 0.490 (±0.016) | 0.372 (±0.100) | 0.420 (±0.018) | 0.561 (±0.022) | **0.865** (±0.010) |
| 1st-Place Count↑ | 0 | 0 | 0 | 2 | 0 | 0 | 0 | 3 | **15** |
| Avg. Rank↓ | 3.65 | 6.65 | 7.00 | 2.95 | 6.70 | 6.15 | 6.20 | 3.55 | **1.95** |
| Avg. AUC↑ | 0.615 | 0.489 | 0.487 | 0.652 | 0.480 | 0.530 | 0.528 | 0.616 | **0.734** |

Table 12: MAE results for the Edge Count prediction task ( regression ). Best results are in **bold**, second best are underlined.

| | Single Model on Individual Networks | | | | | | | Transfer Models |
|---|---|---|---|---|---|---|---|---|
| Dataset | HTGN | TGCN | GCLSTM | ROLAND | EGCN | GraphPulse | WinGNN | Hydra (Ours) |
| MIR | 0.059 (±0.007) | 0.044 (±0.009) | 0.047 (±0.003) | 0.039 (±0.000) | 0.057 (±0.015) | 0.059 (±0.001) | 0.046 (±0.010) | **0.016** (±0.005) |
| DOGE2.0 | 0.101 (±0.035) | 0.063 (±0.017) | 0.106 (±0.029) | 0.052 (±0.003) | 0.092 (±0.031) | 0.046 (±0.000) | **0.045** (±0.003) | 0.187 (±0.008) |
| MUTE | 0.025 (±0.006) | 0.038 (±0.004) | **0.017** (±0.001) | 0.040 (±0.007) | 0.049 (±0.005) | 0.025 (±0.002) | 0.027 (±0.003) | 0.021 (±0.022) |
| EVERMOON | **0.010** (±0.001) | 0.021 (±0.013) | 0.025 (±0.007) | 0.016 (±0.016) | 0.030 (±0.010) | 0.235 (±0.005) | 0.025 (±0.004) | 0.017 (±0.000) |
| DERC | 0.038 (±0.015) | 0.059 (±0.011) | **0.016** (±0.005) | 0.060 (±0.008) | 0.023 (±0.007) | 0.023 (±0.003) | 0.032 (±0.001) | 0.021 (±0.003) |
| ADX | 0.017 (±0.001) | 0.018 (±0.003) | **0.016** (±0.001) | 0.017 (±0.002) | 0.021 (±0.002) | 0.019 (±0.001) | **0.016** (±0.000) | 0.025 (±0.008) |
| HOICHI | 0.034 (±0.010) | **0.020** (±0.001) | 0.046 (±0.013) | 0.020 (±0.003) | 0.034 (±0.013) | 0.044 (±0.002) | 0.028 (±0.005) | 0.027 (±0.020) |
| SDEX | 0.080 (±0.029) | 0.128 (±0.046) | **0.058** (±0.007) | 0.121 (±0.002) | 0.085 (±0.025) | 0.106 (±0.005) | 0.128 (±0.008) | 0.095 (±0.043) |
| BAG | **0.022** (±0.003) | 0.023 (±0.017) | 0.025 (±0.002) | 0.030 (±0.016) | 0.027 (±0.005) | 0.063 (±0.001) | 0.260 (±0.064) | 0.041 (±0.020) |
| XCN | 0.074 (±0.006) | 0.112 (±0.042) | 0.107 (±0.024) | 0.120 (±0.035) | 0.121 (±0.011) | 0.118 (±0.000) | 0.072 (±0.018) | **0.062** (±0.015) |
| ETH2x-FLI | 0.055 (±0.015) | 0.079 (±0.019) | 0.177 (±0.057) | 0.040 (±0.026) | 0.066 (±0.016) | 0.144 (±0.009) | 0.030 (±0.011) | **0.023** (±0.001) |
| stkAAVE | 0.083 (±0.008) | 0.087 (±0.004) | 0.092 (±0.017) | 0.104 (±0.012) | 0.079 (±0.010) | 0.096 (±0.006) | 0.100 (±0.004) | **0.051** (±0.022) |
| GLM | 0.072 (±0.010) | 0.058 (±0.005) | 0.063 (±0.002) | 0.058 (±0.003) | 0.060 (±0.001) | 0.076 (±0.001) | **0.054** (±0.003) | 0.087 (±0.008) |
| QOM | 0.042 (±0.005) | 0.085 (±0.020) | 0.053 (±0.030) | 0.057 (±0.030) | 0.069 (±0.013) | 0.055 (±0.001) | 0.046 (±0.006) | **0.036** (±0.013) |
| WOJAK | 0.009 (±0.002) | 0.012 (±0.003) | 0.013 (±0.004) | 0.016 (±0.008) | 0.013 (±0.007) | 0.057 (±0.008) | **0.006** (±0.002) | 0.010 (±0.004) |
| DINO | 0.069 (±0.020) | 0.025 (±0.008) | 0.040 (±0.002) | **0.014** (±0.004) | 0.039 (±0.011) | 0.087 (±0.002) | 0.021 (±0.008) | 0.017 (±0.007) |
| Metis | 0.038 (±0.002) | 0.054 (±0.001) | 0.047 (±0.003) | 0.057 (±0.008) | 0.053 (±0.004) | 0.066 (±0.006) | 0.043 (±0.001) | **0.034** (±0.010) |
| REPv2 | 0.117 (±0.013) | 0.108 (±0.004) | 0.115 (±0.004) | **0.106** (±0.001) | 0.128 (±0.036) | 0.119 (±0.001) | 0.118 (±0.001) | 0.111 (±0.008) |
| TRAC | 0.026 (±0.004) | 0.036 (±0.010) | 0.061 (±0.006) | 0.023 (±0.004) | 0.036 (±0.003) | **0.017** (±0.000) | 0.040 (±0.014) | 0.021 (±0.001) |
| BEPRO | 0.009 (±0.001) | 0.009 (±0.003) | 0.009 (±0.002) | 0.015 (±0.017) | **0.007** (±0.001) | **0.007** (±0.000) | **0.007** (±0.002) | 0.011 (±0.008) |
| 1st-Place Count↑ | 2 | 1 | 4 | 3 | 1 | 2 | 5 | **6** |
| Avg. Rank↓ | 3.95 | 4.60 | 4.58 | 4.58 | 5.30 | 5.72 | 3.93 | **3.35** |
| Avg. MAE↓ | 0.049 | 0.054 | 0.057 | 0.050 | 0.054 | 0.073 | 0.057 | **0.046** |

Table 13: MAE results for the New Node Count prediction task ( regression ). Best results are in **bold**, second best are underlined.

| Dataset | Single Model on Individual Networks | | | | | | | Transfer Models |
|---|---|---|---|---|---|---|---|---|
| | **HTGN** | **TGCN** | **GCLSTM** | **ROLAND** | **EGCN** | **GraphPulse** | **WinGNN** | Hydra (**ours**) |
| MIR | 0.031 (±0.002) | 0.028 (±0.001) | 0.039 (±0.005) | 0.025 (±0.018) | 0.030 (±0.001) | 0.025 (±0.001) | 0.037 (±0.011) | **0.013** (**±0.004**) |
| DOGE2.0 | 0.046 (±0.009) | 0.064 (±0.021) | **0.044** (**±0.014**) | 0.073 (±0.019) | 0.051 (±0.031) | 0.157 (±0.010) | 0.098 (±0.009) | 0.092 (±0.008) |
| MUTE | **0.021** (**±0.003**) | 0.041 (±0.005) | 0.030 (±0.003) | 0.048 (±0.002) | 0.053 (±0.003) | 0.031 (±0.001) | 0.051 (±0.015) | 0.025 (±0.005) |
| EVERMOON | **0.010** (**±0.004**) | 0.017 (±0.008) | 0.026 (±0.009) | 0.029 (±0.019) | 0.028 (±0.017) | 0.222 (±0.005) | 0.022 (±0.002) | 0.012 (±0.005) |
| DERC | 0.028 (±0.009) | 0.034 (±0.018) | 0.016 (±0.024) | 0.043 (±0.024) | 0.019 (±0.004) | 0.043 (±0.011) | 0.027 (±0.003) | **0.015** (**±0.004**) |
| ADX | 0.014 (±0.001) | **0.010** (**±0.001**) | 0.011 (±0.001) | 0.024 (±0.296) | 0.012 (±0.000) | 0.024 (±0.002) | 0.011 (±0.000) | 0.016 (±0.005) |
| HOICHI | 0.044 (±0.004) | 0.035 (±0.025) | 0.053 (±0.028) | 0.204 (±0.028) | 0.039 (±0.023) | 0.066 (±0.001) | **0.027** (**±0.004**) | 0.029 (±0.008) |
| SDEX | 0.075 (±0.002) | 0.093 (±0.006) | 0.069 (±0.002) | 0.088 (±0.002) | 0.077 (±0.008) | 0.087 (±0.002) | 0.090 (±0.025) | **0.063** (**±0.006**) |
| BAG | 0.023 (±0.007) | 0.031 (±0.008) | **0.019** (**±0.007**) | 0.053 (±0.006) | 0.030 (±0.004) | 0.054 (±0.000) | 0.230 (±0.062) | 0.052 (±0.002) |
| XCN | 0.014 (±0.001) | 0.015 (±0.006) | 0.017 (±0.006) | 0.012 (±0.012) | 0.017 (±0.007) | 0.040 (±0.001) | 0.015 (±0.002) | **0.009** (**±0.002**) |
| ETH2x-FLI | 0.031 (±0.010) | 0.041 (±0.001) | 0.069 (±0.010) | **0.020** (**±0.004**) | 0.030 (±0.002) | 0.092 (±0.002) | 0.028 (±0.003) | 0.030 (±0.004) |
| stkAAVE | 0.128 (±0.018) | 0.136 (±0.008) | 0.128 (±0.018) | 0.154 (±0.005) | 0.124 (±0.005) | 0.151 (±0.000) | 0.147 (±0.008) | **0.078** (**±0.017**) |
| GLM | **0.066** (**±0.002**) | 0.068 (±0.001) | 0.068 (±0.008) | 0.068 (±0.002) | 0.067 (±0.015) | 0.092 (±0.000) | **0.066** (**±0.006**) | 0.094 (±0.000) |
| QOM | 0.035 (±0.006) | 0.038 (±0.010) | 0.032 (±0.020) | 0.018 (±0.010) | 0.035 (±0.020) | 0.033 (±0.004) | 0.029 (±0.007) | **0.012** (**±0.001**) |
| WOJAK | 0.008 (±0.001) | 0.009 (±0.001) | 0.015 (±0.003) | 0.029 (±0.017) | 0.014 (±0.003) | 0.067 (±0.005) | **0.007** (**±0.001**) | 0.009 (±0.000) |
| DINO | 0.061 (±0.011) | 0.024 (±0.002) | 0.028 (±0.009) | **0.013** (**±0.005**) | 0.030 (±0.005) | 0.085 (±0.001) | 0.051 (±0.029) | 0.018 (±0.007) |
| Metis | **0.034** (**±0.010**) | 0.045 (±0.006) | 0.041 (±0.006) | 0.043 (±0.005) | 0.042 (±0.004) | 0.054 (±0.001) | **0.034** (**±0.002**) | **0.034** (**±0.010**) |
| REPv2 | 0.061 (±0.003) | 0.061 (±0.004) | 0.075 (±0.003) | **0.055** (**±0.002**) | 0.063 (±0.004) | 0.068 (±0.000) | 0.063 (±0.000) | 0.066 (±0.004) |
| TRAC | 0.043 (±0.021) | 0.030 (±0.003) | 0.071 (±0.021) | 0.021 (±0.006) | 0.025 (±0.009) | **0.018** (**±0.000**) | 0.044 (±0.010) | 0.022 (±0.005) |
| BEPRO | 0.012 (±0.002) | 0.010 (±0.002) | 0.011 (±0.017) | 0.011 (±0.011) | 0.009 (±0.000) | 0.012 (±0.000) | 0.006 (±0.001) | **0.004** (**±0.001**) |
| 1st-Place Count↑ | 4 | 1 | 2 | 3 | 0 | 1 | 4 | **8** |
| Avg. Rank ↓ | 3.45 | 4.10 | 4.00 | 4.35 | 3.90 | 5.75 | 4.05 | **2.50** |
| Avg. MAE ↓ | 0.039 | 0.042 | 0.043 | 0.052 | 0.040 | 0.071 | 0.054 | **0.035** |

Table 14: MAE results for the Influential Node Count prediction task ( regression ). Best results are in **bold**, second best are underlined.

| Dataset | Single Model on Individual Networks | | | | | | | Transfer Models |
|---|---|---|---|---|---|---|---|---|
| | **HTGN** | **TGCN** | **GCLSTM** | **ROLAND** | **EGCN** | **GraphPulse** | **WinGNN** | Hydra (**ours**) |
| MIR | 0.114 (±0.003) | 0.119 (±0.005) | 0.105 (±0.020) | 0.115 (±0.020) | 0.114 (±0.017) | 0.127 (±0.002) | 0.082 (±0.007) | **0.039** (**±0.005**) |
| DOGE2.0 | 0.064 (±0.031) | 0.090 (±0.023) | **0.053** (**±0.021**) | 0.070 (±0.015) | 0.059 (±0.021) | 0.087 (±0.004) | 0.091 (±0.009) | 0.126 (±0.032) |
| MUTE | 0.028 (±0.004) | 0.042 (±0.002) | **0.018** (**±0.002**) | 0.051 (±0.006) | 0.042 (±0.012) | 0.021 (±0.001) | 0.045 (±0.033) | 0.045 (±0.027) |
| EVERMOON | **0.011** (**±0.002**) | 0.014 (±0.006) | 0.018 (±0.008) | 0.014 (±0.004) | 0.032 (±0.015) | 0.235 (±0.004) | 0.026 (±0.006) | 0.038 (±0.024) |
| DERC | 0.069 (±0.007) | 0.084 (±0.001) | 0.053 (±0.002) | 0.104 (±0.011) | 0.058 (±0.010) | 0.048 (±0.001) | 0.077 (±0.003) | **0.033** (**±0.017**) |
| ADX | 0.016 (±0.003) | 0.015 (±0.000) | **0.012** (**±0.001**) | 0.022 (±0.009) | 0.015 (±0.001) | 0.020 (±0.001) | 0.015 (±0.004) | 0.030 (±0.001) |
| HOICHI | 0.039 (±0.004) | **0.020** (**±0.007**) | 0.046 (±0.017) | 0.030 (±0.015) | 0.034 (±0.016) | 0.047 (±0.009) | 0.024 (±0.016) | 0.055 (±0.028) |
| SDEX | 0.037 (±0.011) | 0.049 (±0.022) | **0.031** (**±0.012**) | 0.058 (±0.017) | 0.037 (±0.005) | 0.067 (±0.009) | 0.066 (±0.025) | 0.042 (±0.019) |
| BAG | **0.009** (**±0.000**) | 0.017 (±0.002) | 0.030 (±0.008) | 0.019 (±0.010) | 0.030 (±0.009) | 0.064 (±0.001) | 0.074 (±0.094) | 0.075 (±0.027) |
| XCN | 0.061 (±0.008) | 0.155 (±0.009) | **0.055** (**±0.003**) | 0.124 (±0.005) | 0.066 (±0.008) | 0.080 (±0.004) | 0.159 (±0.029) | 0.126 (±0.032) |
| ETH2x-FLI | 0.050 (±0.007) | 0.053 (±0.035) | 0.100 (±0.014) | **0.023** (**±0.005**) | 0.044 (±0.010) | 0.108 (±0.003) | 0.033 (±0.010) | 0.033 (±0.020) |
| stkAAVE | 0.062 (±0.001) | 0.080 (±0.009) | 0.062 (±0.001) | 0.057 (±0.005) | 0.061 (±0.006) | 0.108 (±0.009) | 0.066 (±0.003) | **0.042** (**±0.013**) |
| GLM | 0.066 (±0.002) | 0.057 (±0.001) | 0.067 (±0.004) | 0.057 (±0.005) | 0.054 (±0.002) | 0.082 (±0.000) | **0.053** (**±0.002**) | 0.075 (±0.017) |
| QOM | 0.060 (±0.008) | 0.088 (±0.040) | 0.061 (±0.032) | 0.074 (±0.018) | 0.075 (±0.015) | 0.076 (±0.004) | 0.062 (±0.012) | **0.039** (**±0.020**) |
| WOJAK | **0.007** (**±0.001**) | 0.008 (±0.003) | 0.012 (±0.003) | 0.027 (±0.010) | 0.011 (±0.004) | 0.066 (±0.019) | **0.007** (**±0.001**) | 0.036 (±0.027) |
| DINO | 0.078 (±0.036) | 0.024 (±0.004) | 0.068 (±0.001) | 0.024 (±0.003) | 0.036 (±0.007) | 0.087 (±0.003) | **0.022** (**±0.000**) | 0.034 (±0.002) |
| Metis | 0.056 (±0.006) | 0.078 (±0.004) | 0.068 (±0.005) | 0.085 (±0.030) | 0.073 (±0.004) | 0.071 (±0.010) | 0.063 (±0.005) | **0.046** (**±0.008**) |
| REPv2 | **0.115** (**±0.000**) | 0.118 (±0.006) | 0.119 (±0.002) | **0.115** (**±0.001**) | 0.138 (±0.027) | 0.127 (±0.000) | 0.126 (±0.001) | 0.129 (±0.022) |
| TRAC | 0.045 (±0.005) | 0.058 (±0.002) | 0.079 (±0.002) | 0.055 (±0.009) | 0.056 (±0.005) | **0.033** (**±0.005**) | 0.074 (±0.009) | 0.043 (±0.023) |
| BEPRO | 0.016 (±0.006) | 0.014 (±0.003) | 0.012 (±0.002) | 0.013 (±0.001) | 0.012 (±0.002) | 0.012 (±0.000) | **0.010** (**±0.001**) | 0.033 (±0.018) |
| 1st-Place Count↑ | 4 | 1 | 5 | 2 | 0 | 1 | 4 | 5 |
| Avg. Rank ↓ | **3.52** | 4.85 | 3.80 | 4.55 | 4.20 | 5.90 | 4.28 | 4.90 |
| Avg. MAE ↓ | **0.050** | 0.059 | 0.053 | 0.057 | 0.052 | 0.078 | 0.059 | 0.056 |

