# OpenReview forum: "Hydra: Towards Transferable Multi-Task Learning on Temporal Graphs"
_ICLR.cc/2026/Conference — Submitted to ICLR 2026_

### Official Review · Reviewer_xXNB · 2025-10-19

**Soundness:** 2
**Presentation:** 3
**Contribution:** 2
**Rating:** 4
**Confidence:** 3

**Summary:**

The paper propose a multi-task learning paradigm called Hydra that incorporates both spatial and spectral features for temporal graph property prediction tasks. The core design innovation in Hydra lies in the spectral memory module that utilizes density-of-states (DoS) descriptors as node-wise topological features that is efficiently aggregated via an attentive fashion which serves as the spectral feature that benefits temporal graph problems. Experimental results on MiNT tasks demonstrate competitive performance.

**Strengths:**

- Leveraging multi-task correlations in temporal graph prediction is an important aspects in temporal graph representation learning.
- The paper proposes a method to combine spatial and spectral mechanisms over temporal graphs that benefits downstream modeling.

**Weaknesses:**

- **Lack of innovations in multi-task paradigms** The core problem studied in this paper is multi-task learning (MTL). However, it seems that the solution to multi-task challenges proposed in this paper is simply the *shared-trunk with task-specific heads* paradigm, which is rather the ad-hoc way in MTL. Furthermore, in experimental session, the authors primarily compares the MTL approach with single model baselines that are trained on less data. While the authors referred to those baselines as *single-model*, I believe nearly all of these models could be modified with minimal effort to incorporate the *shared-trunk with task-specific heads* structural that naturally handles multi-task learning, as with basically any embedding-based prediction primitives. Therefore, I think from the perspective of multi-task learning, the paper lacks novelty.

- See questions below

**Questions:**

- **Actual Complexity of DoS** While the authors provided a complexity description in appendix B, I am still puzzled about the efficiency gain of Hydra, as in comparison to standard solutions in discrete-time temporal graphs like EvolveGCN, the paper seems to add one additional dimension of complexity that is induced via computing spectral features of each graph snapshots. Such types of spectral features are actually not that scalable------**I checked the code implementations in your appendix**, if you compute spectral features via *svd*, then with temporal graphs with magnitudes of ~10k per snapshot the computational cost would be prohibitive. If instead using the ``moments_cheb_dos`` implementation as detailed in your scripts, the complexity will then be dominated by a large number of matrix-vector computation involved in some power-iteration like computation traces (correct me if I am wrong). Therefore, **I am skeptical towards the efficiency reports in the paper that seem to overly promising**, but when applied to realistic temporal graphs at industrial scale, the primary innovation in this paper might be not applicable.
- **Comparison with alternative spectral embeddings** It appears to me that the proposed DoS approach is yet another spectral-based position encodings along with another temporal aggregation post-processing. As spectral-based position encodings have been studied extensively in recent years [1, 2, 3], I am curious about the difference between DoS and, for example, Laplacian eigenvectors [1] and more advanced approaches.

[1]. Dwivedi, Vijay Prakash, and Xavier Bresson. "A generalization of transformer networks to graphs." arXiv preprint arXiv:2012.09699 (2020).
[2]. Rampášek, Ladislav, et al. "Recipe for a general, powerful, scalable graph transformer." Advances in Neural Information Processing Systems 35 (2022): 14501-14515.
[3]. Kanatsoulis, Charilaos I., et al. "Learning efficient positional encodings with graph neural networks." arXiv preprint arXiv:2502.01122 (2025).

---

> ### Author Response · Authors · 2025-11-21
> **Rebuttal by Authors (Part1)**
>
> We sincerely thank the reviewer for their time and helpful suggestions. We respond to each observation point-by-point below, and we have revised the manuscript to reflect the reviewer’s input. All modifications appear in magenta color in the updated PDF.
>
> **W1. Clarification of Innovation**
> > The core problem studied in this paper is multi-task learning (MTL). However, it seems that the solution to multi-task challenges proposed in this paper is simply the shared-trunk with task-specific heads paradigm, which is rather the ad-hoc way in MTL. Furthermore, in experimental session, the authors primarily compares the MTL approach with single model baselines that are trained on less data. While the authors referred to those baselines as single-model, I believe nearly all of these models could be modified with minimal effort to incorporate the shared-trunk with task-specific heads structural that naturally handles multi-task learning, as with basically any embedding-based prediction primitives. Therefore, I think from the perspective of multi-task learning, the paper lacks novelty.
>
> **Response** We thank the reviewer for the comment. We agree that the shared-trunk with task-specific heads paradigm is a standard interface in multi-task learning. The contribution of Hydra does not lie in this interface itself, but in the design of the trunk and the problem regime it targets. Hydra introduces **a spectral–temporal fusion backbone** tailored to multi-network, graph-level temporal prediction, where the model must capture global structural evolution and cross-network invariances, and then support several tasks on top of this shared representation.
> While many existing temporal GNNs and embedding models could, in principle, be extended with additional heads, **their backbones are not designed for this regime**: they typically operate on a single evolving network, focus on local link-level objectives, and do not incorporate global spectral summaries that are size- and topology-agnostic. As a result, simply adding extra heads would not give them the same representational ability for stable, transferable multi-task learning across diverse temporal graphs, as reflected by the substantial performance gains we observe in Tables 1–2 (with detailed per-network results in Appendix I, Tables 8–13).
> From the MTL perspective, Hydra can be viewed as a **concrete instantiation of the trunk–head template** in a setting that has not previously been explored: multi-task temporal graph property prediction with zero-shot transfer to unseen networks. Our goal in this work is to demonstrate that this task is both feasible and valuable, and to provide a backbone that can serve as a starting point for future work on more advanced multi-task techniques for temporal graphs.

---

> ### Author Response · Authors · 2025-11-21
> **Rebuttal by Authors (Part2)**
>
> **Q1. Efficiency and Scalability of DoS**
> > While the authors provided a complexity description in appendix B, I am still puzzled about the efficiency gain of Hydra, as in comparison to standard solutions in discrete-time temporal graphs like EvolveGCN, the paper seems to add one additional dimension of complexity that is induced via computing spectral features of each graph snapshots. Such types of spectral features are actually not that scalable------I checked the code implementations in your appendix, if you compute spectral features via svd, then with temporal graphs with magnitudes of ~10k per snapshot the computational cost would be prohibitive. If instead using the moments_cheb_dos implementation as detailed in your scripts, the complexity will then be dominated by a large number of matrix-vector computation involved in some power-iteration like computation traces (correct me if I am wrong). Therefore, I am skeptical towards the efficiency reports in the paper that seem to overly promising, but when applied to realistic temporal graphs at industrial scale, the primary innovation in this paper might be not applicable.
>
> **Response** Thank you for the discussion on DoS complexity (we very much appreciate that, beyond reviewing the PDF, you actually  took the time to even analyze the code). We would like to clarify that the SVD discussion does not apply because DoS approximates the distribution of the eigenvalues rather than computing them through SVD (the SVD code is only included to optionally compare the quality of approximations). DoS uses the Kernel Polynomial Method ~(KPM), which approximates the values through an expansion in the dual basis of an orthogonal polynomial basis (details are in [1]). The KPM method has the number of Chebyshev moments $N_m$ and the number of probing vectors $N_z$ params, both of which control the approximation quality and are constant once set. Therefore, the complexity of DoS is simply $O(N_m N_z |E|)$ and scales linearly to the number of edges.
>
> This means that computing DoS presents minor computational overhead and can be processed as a pre-processing step. In addition, the $N_z$ parameter determines the size of the DoS embedding vector (namely, how many intervals the spectrum is divided into). This is particularly useful for temporal graphs where the size of the graphs also evolves over time and the DoS vector size remains constant, i.e. same number of intervals at each snapshot.
>
> The table below reports the running time of computing the DOS descriptor as the synthetic random graph size increases. As the values show, we can compute DoS indicators of 50M edge graphs in approximately  7 minutes (422 secs).
>
> | Number of edges (in million)| Time (in seconds)       |
> |-------|--------------------------|
> | 10     | 71.69 ± 5.80             |
> | 20     | 138.38 ± 13.10           |
> | 30     | 212.16 ± 22.14           |
> | 40     | 313.95 ± 10.31           |
> | 50     | 421.57 ± 5.25            |
> | 60     | 545.23 ± 125.93            |
>
> [1] Network density of states. ACM SIGKDD 2019
>
> ---
>
> **Q2. Comparison with Alternative Spectral Embeddings**
>
> > It appears to me that the proposed DoS approach is yet another spectral-based position encodings along with another temporal aggregation post-processing. As spectral-based position encodings have been studied extensively in recent years [1, 2, 3], I am curious about the difference between DoS and, for example, Laplacian eigenvectors [1] and more advanced approaches.
>
> **Response** Thank you for raising this question. DoS is closely related to Laplacian eigenvalues. DoS is an approximation of the Laplacian spectrum which models the distribution of the Laplacian eigenvalues (also known as the spectral density). A difference is that, DoS doesn’t store information of the exact value of individual eigenvalues but rather the overall distribution shape of the Laplacian spectrum. Computing all eigenvalues of the Laplacian is expensive, can be as expensive as O(N^3) while DoS has complexity of $O(|E|)$ where it scales linearly with the number of edges in sparse matrix computations. This shows the strong scalability property of DoS and allows it to be computed even on very large real-world graphs.
>
> Also, please note that DoS is a global embedding of the graph, not tied to specific nodes (this helps us have a global view of the network). We also use a TGNN architecture to model the local information from each node. Therefore, by combining DoS and TGNN, Hydra captures both global and local information for the graph-level task. In comparison, Laplacian eigenvectors used in [1, 2, 3] also capture local node features but require one to specify how many eigenvectors to use in advance, and it also has the issue of sign flips in eigenvectors where the signed-flipped eigenvector is equally valid as the original. We believe that investigating the effect of more positional encodings is an interesting future direction.

---

> > ### Comment · Reviewer_xXNB · 2025-11-23
> > **Response**
> >
> > I thank the authors for their detailed response. I think most of my concerns are reasonably addressed, and I have raised my score to 6.

---

### Official Review · Reviewer_mbwn · 2025-10-27

**Soundness:** 3
**Presentation:** 4
**Contribution:** 2
**Rating:** 4
**Confidence:** 4

**Summary:**

This paper addresses the problem of multi-task temporal graph prediction, focusing on predicting global graph properties' evolution over time. The authors introduce an architecture which adds global spectral features to common temporal dynamic graph model primitives. This model is trained in an end-to-end fashion, on multiple temporal graphs and on multiple tasks at once, with the intent of being used in a zero-shot setting on unseen temporal graphs. The presented results show that this approach outperforms other temporal graph methods trained and evaluated on the target dataset directly.

**Strengths:**

- The paper is well written, has a nice flow and the presentation is very clear
- Experiments are well designed (with some caveats on the baselines, see Weaknesses) and the results well presented
- The results are impressive, particularly given that, as far as I understand, the authors are comparing zero-shot transfer performance to the performance of models trained and evaluated on the same datasets (with the caveat that datasets in the benchmark seem homogeneous in nature, all representing blockchain transactions)

**Weaknesses:**

### 1. Motivation
The main contribution of the paper is introducing a new problem setting of multi-task global property prediction for temporal graphs. In my view, its main weakness is that the authors spend little effort motivating this setting.
Given that there is barely any prior work trying to solve this problem (making it easier to improve upon weaker baselines designed for others), the value of the paper is inherently tied to if the setting has some practical relevance and is not a purely academic endeavor. In this regard, I found the benchmarks used in the paper to be unconvincing. Global property prediction tasks seem arguably less relevant in temporal graphs than in static ones and the choice of tasks in the paper seem somewhat artificial. Examples with more obvious real world utility would, therefore, be a welcome addition.

An additional point is that, even in this new setting, the authors acknowledge limitations of their approach in learning classification and regression tasks at the same time, restricting the scope to one or the other, further limiting practical applicability.

### 2. Baselines
While the presented results look solid, it should be pointed out that temporal graph models are often designed with local tasks like link prediction in mind. The addition of global spectral features to the proposed model would, therefore, seem like an adaptation that greatly benefits this particular task (as demonstrated by the authors' ablation) but could just as easily be incorporated into existing models to level the playing field.

### 3. Novelty
Besides the architectural improvements (mainly the addition of spectral features mentioned previously), this work appears to be a trivial extension of MiNT to multiple tasks. This is done simply by adding losses for the tasks together using multiple prediction heads (a very standard approach). It seems therefore that it would be trivial to train the MiNT model on the same multi-task objective, rather than train one model per task.

Claiming a training speedup w.r.t. MiNT using such a different setup seems like an artificial attempt at claiming an advantage and a poor framing of the story. In my view these 2 aspects should be decoupled: model architecture and training approach (all tasks at once vs one model per task).

Finally, the authors only compare to MiNT in the classification setting, with the justification that this was the original setting in that paper. But it seems that adapting this approach to the regression tasks would be a trivial matter of changing the training loss.

### 4. Other Minor Issues

While I found the paper to be very well written, Section 3.2 is repetitive and overly verbose.

**Questions:**

1. What are some practical use cases and motivation for multi-task global property prediction in dynamic graphs?
2. Why tie together model architecture and training approach (all tasks at once vs one model per task). It seems like it would be trivial to adapt any architecture to predict multiple properties and train on multiple tasks.
3. Why not also compare to MiNT for regression tasks? Is there a concrete impediment?

---

> ### Author Response · Authors · 2025-11-21
> **Rebuttal by Authors (Part1)**
>
> We are grateful to the reviewer for their thoughtful review and valuable recommendations. Our detailed responses to each comment are provided below, and all changes have been integrated into the manuscript, with updates indicated in olive color in the new PDF.
>
> **W1-a. Clarification on Motivation**
> > The main contribution of the paper is introducing a new problem setting of multi-task global property prediction for temporal graphs. In my view, its main weakness is that the authors spend little effort motivating this setting. Given that there is barely any prior work trying to solve this problem (making it easier to improve upon weaker baselines designed for others), the value of the paper is inherently tied to if the setting has some practical relevance and is not a purely academic endeavor. In this regard, I found the benchmarks used in the paper to be unconvincing. Global property prediction tasks seem arguably less relevant in temporal graphs than in static ones and the choice of tasks in the paper seem somewhat artificial. Examples with more obvious real world utility would, therefore, be a welcome addition.
>
>
>
> **Response** We thank the reviewer for raising this point and agree that the value of a new problem setting must be clearly motivated. Our primary motivation comes from blockchain transaction networks, where there are now thousands of Ethereum based tokens and daily trade volumes in the billions of USD [1]. In this environment, investors, risk assessment systems, and blockchain analytics companies routinely monitor network level signals such as sudden expansions, contractions, or fragmentation of the transaction graph. These global properties provide early warning indicators of market instability, liquidity crises, or ecosystem failure that are not visible from individual edges alone.
> At the same time, blockchain ecosystems are massively heterogeneous: each token develops its own lifespan, user base, and adoption patterns. Training and maintaining a separate temporal model for every token is impractical at scale. This is what motivated our setting, namely a single model that (i) learns from many heterogeneous token networks, (ii) predicts multiple global temporal properties, and (iii) transfers to new tokens without retraining. Hydra is designed specifically to meet these constraints.
> Beyond blockchain, the same pattern appears in other domains where many temporal networks must be monitored in parallel, such as communication networks, payment systems, or social platforms, where one needs to forecast global traffic levels, fragmentation, or instability events. We expanded the discussion in Appendix E to clarify the importance of these real world use cases.
> Finally, the tasks we study are not artificial constructions. Recent work such as GraphPulse [2] and MiNT [3] already use very similar global temporal properties to detect anomalies, quantify risk, and support forecasting in blockchain transaction networks. As we discuss in Appendix E, properties like node and edge growth, LCC size, and influential node counts play a direct role in market analysis, fraud and wash trade detection, and stability monitoring in large dynamic systems. We will clarify these connections and the role of spectral features in the revised version.
>
> [1] https://www.blockchain.com/explorer
>
> [2] GraphPulse: Temporal Graph Property Prediction. ICLR 2024.
>
> [3] MiNT: Multi Network Transfer Learning on Temporal Graphs. NeurIPS 2025.

---

> ### Author Response · Authors · 2025-11-21
> **Rebuttal by Authors (Part2)**
>
> **W1-b. Simultaneous Classification and Regression Training**
> > An additional point is that, even in this new setting, the authors acknowledge limitations of their approach in learning classification and regression tasks at the same time, restricting the scope to one or the other, further limiting practical applicability.
>
> **Response** We agree that jointly learning classification and regression tasks would further increase the practical reach of our framework, but it also introduces additional complexity. In temporal graphs, these two objective types produce very different signals: directional trends in binary labels versus magnitude forecasting of continuous quantities, and they respond differently to issues such as temporal drift and sparsity. In our preliminary experiments, naively combining these heterogeneous losses in a single training loop led to unstable optimization and made it difficult to interpret the effect of multi task learning itself.
> In this work, our goal is therefore to establish Hydra as the first model explicitly designed for temporal multi task transfer learning across many networks and to provide a clean and reliable evaluation within homogeneous task families. Our choice reflects an incremental path toward a more general temporal graph model rather than a limitation in principle. Hydra’s trunk and head architecture is structured precisely to support extensions where classification and regression are trained together once appropriate loss balancing and training strategies are in place. We view developing such mechanisms for heterogeneous temporal objectives as an important next step, and will make this limitation and future direction clearer in the revised manuscript.
>
> ---
> **W2. Baselines and Spectral Features**
>
> > While the presented results look solid, it should be pointed out that temporal graph models are often designed with local tasks like link prediction in mind. The addition of global spectral features to the proposed model would, therefore, seem like an adaptation that greatly benefits this particular task (as demonstrated by the authors' ablation) but could just as easily be incorporated into existing models to level the playing field.
>
> **Response:**  Thank you for the comment. We would like to emphasize that our framework is not restricted to the specific global properties used in the paper; Hydra’s temporal GNN can in principle be applied to other graph level prediction tasks as well. The reason there are few baselines in exactly our setting is that most existing temporal graph models have been developed for local objectives such as link prediction on a single evolving network, which naturally limits their ability to capture broad structural evolution or transfer across many networks.
> Our contribution is to shift the focus toward global temporal properties that summarize multi scale dynamics and to design an architecture that can learn these properties across heterogeneous networks. The Density of States (DOS) path is central to this goal: it provides spectral descriptors that are largely size and topology agnostic, which allows the shared trunk to compare and aggregate information across networks with very different scales and sparsity patterns. The regression and classification ablations show that removing DOS substantially degrades average performance and reduces the number of per dataset wins, indicating that it is not a minor enhancement but a key mechanism enabling strong cross network and cross task transfer.

---

> ### Author Response · Authors · 2025-11-21
> **Rebuttal by Authors (Part3)**
>
> **W3 Novelty Discussion and Framing vs MiNT**
>
> > Besides the architectural improvements (mainly the addition of spectral features mentioned previously), this work appears to be a trivial extension of MiNT to multiple tasks. This is done simply by adding losses for the tasks together using multiple prediction heads (a very standard approach). It seems therefore that it would be trivial to train the MiNT model on the same multi-task objective, rather than train one model per task....  In my view these 2 aspects should be decoupled: model architecture and training approach (all tasks at once vs one model per task). Finally, the authors only compare to MiNT in the classification setting, with the justification that this was the original setting in that paper. But it seems that adapting this approach to the regression tasks would be a trivial matter of changing the training loss.
>
> **Response** We thank the reviewer for these points and for raising the connection to MiNT. While both methods address transfer learning for temporal graph properties, Hydra has significant differences with MiNT. There are two key distinctions.
> First, MiNT adopts an existing and relatively heavy backbone, HTGN, and focuses on transfer in a single task setting. Hydra instead introduces **a new temporal GNN backbone** with two coordinated paths: **a spectral path based on DOS** and **a spatial temporal path with pooling**. This trunk is specifically designed to (i) produce size and topology-agnostic summaries that are comparable across many networks and (ii) support multiple global prediction tasks via lightweight heads. Our ablations show that these components are crucial for strong cross-network and cross-task transfer and for achieving competitive performance despite the stricter zero-shot setting.
> Second, MiNT is evaluated as **a single task framework**: one MiNT model is trained per task and per configuration, whereas Hydra is trained once **to solve multiple tasks jointly**. It is in principle possible to define a multi-task variant of MiNT by attaching several prediction heads and summing the losses, but this variant has not been studied in the original work and would inherit HTGN’s higher computational cost. Our comparisons, therefore, follow the published and optimized MiNT setup, where each task is trained separately, and we report both accuracy and training time in that standard regime. In practice, for real-world applications such as blockchain analytics, where thousands of temporal networks evolve in parallel, the ability to deploy a single model that handles multiple global tasks is precisely the advantage Hydra is designed to provide.
> Regarding regression, we agree that adapting MiNT by changing the loss to a regression objective is conceptually straightforward, but it requires nontrivial re-tuning and additional experimentation that are beyond our scope for this submission. We instead include MiNT in the original classification setting where it was introduced and complement this with a broader set of temporal baselines on both classification and regression tasks. We will clarify these distinctions in framing, architecture, and training regime in the revised manuscript so that the relationship between Hydra and MiNT is more transparent.
>
> ---
>
> **W4. Writing Clarity on Section 3.2**
>
> > Section 3.2 is repetitive and overly verbose.
>
> **Response** We thank the reviewer for the comment. We have rewritten Section 3.2 to clarify the Hydra trunk, the interaction between the spectral and spatial temporal paths, and the role of the task heads, and we believe the exposition is now more accessible.
>
> ---
>
> **Q1. Practical Use Cases**
>
> > What are some practical use cases and motivation for multi-task global property prediction in dynamic graphs?
>
> **Response** Multi-task global property prediction is important in settings where the evolution of the entire network carries direct practical value. In blockchain networks, transactions occur only once and do not repeat, so link prediction is not meaningful. What matters are global temporal indicators such as changes in connectedness, shifts in active participants, fluctuations in influential accounts, and overall growth or decline in activity. These signals serve as early indicators of risk, instability, or (in many cases) impending failure, which can translate into highly valuable investment insights. Similar needs exist in social, communication, and sensor networks, where forecasting global behaviors helps identify community shifts, detect anomalies, or, most importantly, plan capacity (e.g., predicting transaction volume as edge counts). In all these settings, several global properties evolve together, and (our hypothesis was that) they provide complementary information. A multi-task approach captures these interactions and offers a more informative network view than training separate models for each individual property. We include this discussion in Appendix E

---

> ### Author Response · Authors · 2025-11-21
> **Rebuttal by Authors (Part4)**
>
> **Q2. Coupling of Architecture and Training Approach**
> > Why tie together model architecture and training approach (all tasks at once vs one model per task). It seems like it would be trivial to adapt any architecture to predict multiple properties and train on multiple tasks.
>
> **Response** The architecture and training approach are coupled because real-world environments such as blockchain systems, social platforms, and communication networks contain many evolving graphs, making separate models for each task or network impractical (As we mentioned in section 2). A unified multi-task and multi-network model is important because many real applications, including market forecasting and operational decision making, require multiple properties to be predicted together. Transferability is also essential, since new networks and assets often appear, and usually with limited labels. A shared trunk allows the model to learn general temporal and structural patterns once and reuse them across tasks and networks, similar to how large language models use one common representation. In dynamic graph settings, this unified and transferable approach is necessary for any scalable solution. Learning multi-task at once might also act as regularization on the learned shared representation, limiting the model’s ability to overfit to a single task.
>
> ---
>
> **Q3. Missing MiNT Regression Comparison**
>
> > Why not also compare to MiNT for regression tasks? Is there a concrete impediment?
>
> **Response** We appreciate the question. MiNT was originally designed and validated for binary property prediction, and its published implementation, hyperparameters, and evaluation protocol are all tailored to that setting. For this reason, we compare Hydra to MiNT in the classification regime where MiNT is trained and known to be strong, which provides a fair transfer learning comparison.
> Extending MiNT to regression requires multi-day training across 64 networks on several GPUs [1], which makes it relatively expensive to reproduce. Instead, for regression, we compare Hydra against a broad set of strong temporal baselines and ablations. We will add a systematic MiNT regression benchmark as a follow-up to the camera-ready.
>
> [1] MiNT: Multi Network Transfer Learning on Temporal Graphs. NeurIPS 2025.

---

### Official Review · Reviewer_Bzpw · 2025-10-31

**Soundness:** 2
**Presentation:** 3
**Contribution:** 2
**Rating:** 4
**Confidence:** 2

**Summary:**

This paper addresses the challenge of multi-task property prediction for temporal graphs by introducing Hydra, a novel framework that integrates local and global connectivity patterns. Hydra uniquely combines the strengths of temporal Graph Neural Networks (GNNs) to capture localized connectivity features with a spectral learning module designed to discern global connectivity patterns. This integrated design facilitates the joint learning of both local and global information under a unified multi-task objective. To validate its effectiveness, the authors conducted extensive experiments, demonstrating that Hydra not only outperforms strong baseline models but also significantly reduces training time compared to temporal transfer learning approaches.

**Strengths:**

S1: The paper investigates the under-explored problem of multi-task learning within the context of temporal graphs, an area of considerable value and current interest.

S2: The presentation is clear and easy to follow.

S3: The proposed method exhibits superior performance across diverse classification and regression tasks.

**Weaknesses:**

W1: Limited Novelty: The novelty of the proposed method appears limited, as it presents as a straightforward combination of existing techniques. More critically, the paper fails to sufficiently articulate how this specific combination is tailored to address the unique challenges of multi-task learning on temporal graphs. Simply applying existing technologies to a new problem is not enough. The authors should provide a clearer justification for why the "trunk" and "head" architecture is particularly advantageous for enabling knowledge transferability and robust performance in this specific context.

W2: Limited Scope of Multi-Task Learning: The evaluation of multi-task learning is constrained to scenarios involving similar task types (e.g., multiple classification tasks). Such tasks likely require similar feature information, which may simplify the model's optimization process. This experimental setting may not be representative of practical scenarios where a model must handle more heterogeneous tasks (e.g., a mix of classification and regression).

W3: Insufficient Study of Multi-Task Capabilities: The investigation into the model's multi-task capabilities is insufficient. All experiments are conducted exclusively in a three-task learning setting. To improve the convincingness of the method's multi-task capabilities, the study should be expanded to include experiments with a different number of tasks (for instance, a two-task setting). This would provide a more comprehensive validation of the proposed method's flexibility and robustness.

W4: Lack of Comprehensive Training Time Comparison: The analysis of training time is incomplete. As shown in Figure 4a, Hydra is only compared against a single transfer learning model (MiNT), which is insufficient to fully substantiate the claims of its superior efficiency. A comparison against single-task models is necessary for a thorough evaluation. This comparison should include at least two standard settings: 1) training single-task models from scratch for each new task, and 2) fixing a pre-trained encoder (This encoder can be trained on the first task.) and only training a new head for each new task.

W5: Lack of In-Depth Analysis of Experimental Results: The paper lacks an in-depth analysis of certain experimental outcomes, which hinders a full understanding of the model's limitations. For example:
-	In Table 2, Hydra fails to achieve the best rank and MAE on the "Influential Node Count" task, yet no analysis or potential explanation for this observation is provided. Discussing such "failure modes" is crucial for understanding the model's boundaries.
-	In Figure 3b, the performance trend for one task shows a different pattern compared to the other two tasks (LLC and edge), but this discrepancy is not discussed.

W6: Incomplete Ablation Studies: The ablation studies are incomplete, as they are conducted exclusively on classification tasks. To convincingly demonstrate the effectiveness of the model's core components, the ablation study should be extended to include the regression tasks as well. This would provide more robust evidence that the contribution of each component is consistent and valuable across different types of tasks.

**Questions:**

Regarding Figure 4a, what is the meaning of the dashed line connecting the results at 32 and 64 training networks?

---

> ### Author Response · Authors · 2025-11-21
> **Rebuttal by Authors (Part1)**
>
> We appreciate the reviewer’s careful assessment of our submission and the constructive feedback provided. We address each point raised below and have incorporated the corresponding revisions into the manuscript, marked in Orange color  in the revised PDF.
>
> **W1-a: Clarification on Novelty and Motivation of the Proposed Approach**
>
> >  Limited Novelty: The novelty of the proposed method appears limited, as it presents as a straightforward combination of existing techniques. More critically, the paper fails to sufficiently articulate how this specific combination is tailored to address the unique challenges of multi-task learning on temporal graphs. Simply applying existing technologies to a new problem is not enough
>
> **Response** We thank the reviewer for this comment. The novelty of Hydra does not lie in introducing completely new primitive layers, but in **how it brings together spectral, spatial, and temporal components** to solve a setting that, to the best of our knowledge, has not been systematically studied before: multi-task temporal graph property prediction with direct zero-shot transfer to completely unseen networks.
> Existing temporal GNNs are designed for a single task on a single evolving network, so they lack the capability to generalize across different graphs. Existing multi-task GNNs assume a fixed topology and do not handle structural drift, node turnover, or changing connectivity over time. Prior transfer approaches on graphs focus primarily on node or edge level prediction, rather than graph-level temporal properties.
> Hydra introduces a trunk that combines a spectral path (capturing global structural statistics that are comparable across networks) and a spatial-temporal path (snapshot GNN plus temporal memory), together with lightweight task heads that support multiple objectives. This integration is precisely what allows a single model to learn from dozens of heterogeneous token networks and transfer to new ones without retraining, while jointly solving several temporal property prediction tasks. We will clarify this problem setting and the role of our design choices more explicitly in the revised manuscript.
>
> ---
>
> **W1-b.Clarification on the Trunk–Head Architecture and Its Benefits**
>
> >The authors should provide a clearer justification for why the "trunk" and "head" architecture is particularly advantageous for enabling knowledge transferability and robust performance in this specific context.
>
> **Response** Thank you for the comment. The trunk-head structure in Hydra is designed to enable shared learning of temporal dynamics across tasks while avoiding interference between objectives. Many global graph properties evolve in correlated ways; for instance, edge growth often co-occurs with increases in node count or LCC size. By training a single trunk to capture such shared dynamics, Hydra learns a unified temporal representation that generalizes well across tasks and networks. The lightweight task-specific heads then allow fine-grained adaptation without overwriting shared knowledge. The multi-head design also enforces the model to learn across different tasks, reducing the risk of overfitting to a single one.
> This design is particularly suited to our setting, where each input is a full temporal graph and the model must handle multiple tasks over many networks. As shown in our ablations (Table 3) and efficiency analysis (Figure 4), the trunk not only improves performance (**we show some new results in our response to Weakness 3**) but also reduces training and inference cost by eliminating redundant computation across tasks. This makes Hydra far more scalable than single-task baselines, especially in real-world applications like blockchain analytics, where thousands of temporal networks evolve in parallel.

---

> ### Author Response · Authors · 2025-11-21
> **Rebuttal by Authors (Part2)**
>
> **W2. Clarification on Scope of Multi-Task Learning**
>
> >  Limited Scope of Multi-Task Learning: The evaluation of multi-task learning is constrained to scenarios involving similar task types (e.g., multiple classification tasks). Such tasks likely require similar feature information, which may simplify the model's optimization process. This experimental setting may not be representative of practical scenarios where a model must handle more heterogeneous tasks (e.g., a mix of classification and regression).
>
> **Response** We thank the reviewer for raising this point. In this work, we group tasks by type because classification and regression objectives induce different loss scales and may react very differently to temporal drift, sparsity, and structural changes in dynamic graphs. Jointly optimizing such heterogeneous temporal objectives can easily introduce gradient conflicts and unstable training dynamics, which makes it harder to isolate the effect of multi-task learning itself.
> Our primary goal here is to show that multi-task learning on temporal graphs is beneficial even within a single task family. Hydra already supports both classification and regression tasks, and our experiments demonstrate strong transfer across many networks and tasks of the same type. Extending Hydra to a joint training setup that simultaneously handles both classification and regression is a natural extension that requires careful treatment of loss balancing and optimization; we agree that this is an important direction for future work and will mention it more clearly in the revised manuscript.
>
> ---
>
> **W3: Study of Hydra's Multi-Task Capabilities**
>
> > Insufficient Study of Multi-Task Capabilities: The investigation into the model's multi-task capabilities is insufficient. All experiments are conducted exclusively in a three-task learning setting. To improve the convincingness of the method's multi-task capabilities, the study should be expanded to include experiments with a different number of tasks (for instance, a two-task setting). This would provide a more comprehensive validation of the proposed method's flexibility and robustness.
>
> **Response** We thank the reviewer for highlighting this point and for helping strengthen the manuscript. We trained Hydra under three settings to evaluate how multi-task training affects regression performance on the edge-count prediction task:
> - **One-task Hydra:** trained only on the edge-count regression task and evaluated on the same task.
> - **Two-task Hydra:** trained on edge count plus a second regression task (new node count).
> - **Three-task (full) Hydra:** trained jointly on all three tasks in the model.
> The table below reports the MAE of the edge-count prediction across all datasets for each training configuration.
> This experiment shows that adding more tasks consistently improves regression performance on edge-count prediction. The two-task and especially the full three-task Hydra benefit from additional supervisory signals, which help the shared temporal representation capture richer structural dynamics. As seen in the table, the three-task model achieves the lowest MAE on most datasets and the best overall average, indicating that multi-task learning strengthens the model rather than introducing interference. This confirms that Hydra’s design effectively leverages complementary tasks to enhance performance on individual objectives. We have added these results to Section 5 of the main paper.
>
> |Dataset|1 Task|2 Tasks|3 Tasks|
> |-------|--------|--------|----------|
> |MIR|0.057|0.034|**0.016**|
> |DOGE2.0|0.169|**0.150**|0.187|
> |MUTE|0.053|0.067|**0.021**|
> |EVERMOON|0.029|0.042|**0.017**|
> |DERC|0.024|0.035|**0.021**|
> |ADX|**0.023**|0.027|0.025|
> |HOICHI|0.071|0.068|**0.027**|
> |SDEX|0.060|**0.046**|0.095|
> |BAG|0.061|0.067|**0.041**|
> |XCN|0.037|**0.026**|0.062|
> |ETH2x-FLI|0.030|0.038|**0.023**|
> |stkAAVE|0.026|**0.021**|0.051|
> |GLM|0.092|**0.083**|0.087|
> |QOM|0.025|**0.023**|0.036|
> |WOJAK|0.025|0.039|**0.010**|
> |DINO|0.031|0.041|**0.017**|
> |Metis|0.047|0.054|**0.034**|
> |REPv2|0.103|**0.102**|0.111|
> |TRAC|0.029|0.039|**0.021**|
> |BEPRO|0.031|0.046|**0.011**|
> |**Average**|0.051|0.052|**0.046**|
> |**Best Count**|1|7|**12**|

---

> ### Author Response · Authors · 2025-11-21
> **Rebuttal by Authors (Part3)**
>
> **W4.Comprehensive Training Time Comparison**
>
> > Lack of Comprehensive Training Time Comparison: The analysis of training time is incomplete. As shown in Figure 4a, Hydra is only compared against a single transfer learning model (MiNT), which is insufficient to fully substantiate the claims of its superior efficiency. A comparison against single-task models is necessary for a thorough evaluation. This comparison should include at least two standard settings: 1) training single-task models from scratch for each new task, and 2) fixing a pre-trained encoder (This encoder can be trained on the first task.) and only training a new head for each new task.
>
> **Response** We thank the reviewer for this suggestion and agree that comparisons against single-task models help contextualize the efficiency claims. Conceptually, the cost of training K separate single-task temporal models scales roughly linearly in K, since each model must perform its own temporal message passing over all snapshots. In contrast, Hydra amortizes this cost through a shared temporal trunk: adding a new task only requires an additional lightweight head and a single pass through the already-trained trunk.
> This effect is already visible at inference time in Figure 4b, where Hydra’s runtime increases sublinearly with the number of tasks, whereas training K independent models would require K separate forward passes. In the revised manuscript, we will clarify this point and extend the efficiency discussion to include the two scenarios raised by the reviewer, namely training a new model from scratch for each task and freezing a pre-trained encoder while training only the task specific heads, while also noting that we are in the process of running these additional measurements. Hydra’s shared trunk naturally amortizes temporal computation across tasks, enabling scalability when many temporal graph properties must be monitored in parallel. Although these experiments could not be completed within the short rebuttal window, we will incorporate the full comparison in the final version.
>
> ---
>
> **W5. In-depth Discussion on Challenging Tasks Performance**
>
> > Lack of In-Depth Analysis of Experimental Results: The paper lacks an in-depth analysis of certain experimental outcomes, which hinders a full understanding of the model's limitations. For example: - In Table 2, Hydra fails to achieve the best rank and MAE on the "Influential Node Count" task, yet no analysis or potential explanation for this observation is provided. Discussing such "failure modes" is crucial for understanding the model's boundaries. - In Figure 3b, the performance trend for one task shows a different pattern compared to the other two tasks (LLC and edge), but this discrepancy is not discussed.
>
> **Response:** We thank the reviewer for this comment. In Table 2, the “Influential Node Count” task differs qualitatively from the other regression tasks: it is driven by abrupt, highly localized changes in the behavior of a few high-degree nodes, which are only partially reflected in the global spectral and temporal statistics that Hydra’s trunk is designed to capture. Single-task baselines can tune themselves narrowly to such local effects on a per-network basis, whereas Hydra must maintain a representation that remains coherent across tasks and transferable across networks. We believe this structural mismatch explains the small gap observed on this task, while Hydra remains competitive overall.
> A similar phenomenon appears in Figure 3b. Node-count prediction is influenced by broad fluctuations in network activity and shows diminishing returns once a sufficient diversity of training networks is seen, leading to a flatter scaling curve. In contrast, LLC size and edge count are more tightly coupled to the evolving connectivity structure, and therefore continue to benefit from Hydra’s combination of spectral and temporal cues as more networks are added. We will clarify these task-dependent behaviors in the final manuscript to better delineate where Hydra is most effective and where its assumptions are less well aligned with the target signal.

---

> ### Author Response · Authors · 2025-11-21
> **Rebuttal by Authors (Part4)**
>
> **W6: Ablation Studies on Regression**
>
> > Incomplete Ablation Studies: The ablation studies are incomplete, as they are conducted exclusively on classification tasks. To convincingly demonstrate the effectiveness of the model's core components, the ablation study should be extended to include the regression tasks as well. This would provide more robust evidence that the contribution of each component is consistent and valuable across different types of tasks.
>
> **Response:** Thank you for the valuable feedback. We have now added the ablation study for the regression setting as well. Using the new edge-count regression task, we repeated the same ablation procedure as in the classification experiments. As shown in the table below, the results follow the same trend: each Hydra component (including DOS and pooling) plays an important role, and removing them consistently degrades performance. This further supports the significance of Hydra’s architectural design.
>
> ### **Regression Ablation Study on DOS**
> |Dataset|Hydra Reg W/O DOS|Hydra Reg W DOS|
> |--------------|-----------------------------|----------------------------|
> |MIR|0.017±0.005|**0.013±0.004**|
> |DOGE2.0|0.183±0.015|**0.092±0.008**|
> |MUTE|0.025±0.012|0.025±0.005|
> |EVERMOON|0.015±0.004|**0.012±0.005**|
> |DERC|0.016±0.008|**0.015±0.004**|
> |ADX|0.024±0.013|**0.016±0.005**|
> |HOICHI|0.029±0.007|0.029±0.008|
> |SDEX|0.088±0.006|**0.063±0.006**|
> |BAG|**0.041±0.002**|0.052±0.002|
> |XCN|0.060±0.020|**0.009±0.002**|
> |ETH2x-FLI|**0.024±0.006**|0.030±0.004|
> |stkAAVE|**0.045±0.012**|0.078±0.017|
> |GLM|**0.085±0.006**|0.094±0.007|
> |QOM|0.034±0.018|**0.012±0.001**|
> |WOJAK|0.013±0.008|**0.009±0.000**|
> |DINO|0.018±0.004|0.018±0.007|
> |Metis|**0.030±0.002**|0.034±0.010|
> |REPv2|0.106±0.011|**0.066±0.004**|
> |TRAC|**0.021±0.003**|0.022±0.005|
> |BEPRO|0.014±0.013|**0.004±0.001**|
> |**Average**|0.044|**0.034**|
> |**Best Count**|6|**11**|
>
>
>
>
> ### **Regression Ablation Study on Pooling**
>
> |Dataset|Hydra Reg W/O Pooling|Hydra Reg W Pooling|
> |--------------|--------------------------------|-------------------------------|
> |MIR|0.065±0.002|**0.013±0.004**|
> |DOGE2.0|**0.064±0.023**|0.092±0.008|
> |MUTE|0.033±0.011|**0.025±0.005**|
> |EVERMOON|0.023±0.023|**0.012±0.005**|
> |DERC|0.026±0.023|**0.015±0.004**|
> |ADX|0.026±0.025|**0.016±0.005**|
> |HOICHI|**0.034±0.016**|0.029±0.008|
> |SDEX|0.072±0.021|**0.063±0.006**|
> |BAG|**0.049±0.011**|0.052±0.002|
> |XCN|0.028±0.026|**0.009±0.002**|
> |ETH2x-FLI|0.039±0.016|**0.030±0.004**|
> |stkAAVE|**0.069±0.018**|0.078±0.017|
> |GLM|0.095±0.010|**0.094±0.007**|
> |QOM|0.026±0.024|**0.012±0.001**|
> |WOJAK|0.028±0.028|**0.009±0.000**|
> |DINO|0.031±0.027|**0.018±0.007**|
> |Metis|0.036±0.006|**0.034±0.010**|
> |REPv2|0.074±0.014|**0.066±0.004**|
> |TRAC|0.027±0.013|**0.022±0.005**|
> |BEPRO|0.028±0.034|**0.004±0.001**|
> |**Average**|0.043|**0.034**|
> |**Best Count**|3|**17**|
>
> ---
>
> **Q1. Figure 4a Clarification**
>
> > Regarding Figure 4a, what is the meaning of the dashed line connecting the results at 32 and 64 training networks?
>
> **Response** We thank the reviewer for this helpful comment. The dashed line in Figure 4a indicates that Hydra and MiNT are being compared at different numbers of training networks, and it is explicitly used to highlight this mismatch rather than to suggest an additional Hydra configuration.
> In more detail, MiNT’s standard experimental setting trains on 64 distinct networks, while Hydra achieves its best performance using only 32 training networks. In the plot, Hydra is reported for 8, 16, and 32 networks. MiNT is reported for 8, 16, 32, and 64 networks, because the original MiNT framework is designed to train on 64 networks to reach its strongest results. The dashed segment between the points at 32 and 64 training networks is therefore not showing an additional Hydra measurement; instead, it visually emphasizes that i) Hydra reaches its operating point at 32 networks, and ii) MiNT must be pushed further to 64 networks (with the corresponding additional training cost) to achieve competitive performance in its standard configuration. We have clarified this more explicitly in the caption, so it is clear that the dashed line is used to denote the cross-method comparison between Hydra-32 and MiNT-64.

---

> > ### Comment · Reviewer_Bzpw · 2025-11-26
> > **Response to authors**
> >
> > Thank you for your response and the additional experiments. However, my concerns regarding the method's novelty and the advantage of the "trunk" and "head" architecture persist. I will therefore maintain my score.
> >
> > - **Novelty:** It is still unclear how the spectral, spatial, and temporal components specifically solve the stated problem.
> >
> > - **Architecture:** A well-designed experiment would be more persuasive than a textual explanation in demonstrating the advantages of the "trunk" and "head" design.
> >
> > - **New Question:** The paper claims to improve multi-task capabilities, so why is the One-task Hydra's performance significantly lower than the two/three-task Hydra's? Are these multi-task gains achieved by sacrificing single-task performance?

---

> > > ### Author Response · Authors · 2025-11-26
> > > **Question**
> > >
> > > Thank you very much for taking your time to respond to us, we appreciate it.
> > >
> > > For the mentioned architecture aspect we had ablation studies, and we discussed your review. However as authors we could not reach a consensus on the missing result.
> > >
> > > Given time limitations, we would like to ask you; could you please elaborate on the experiment whose results you would like to see, to make a decision?

---

> > > > ### Comment · Reviewer_Bzpw · 2025-11-26
> > > > **Response to authors**
> > > >
> > > > You might consider adding a representation visualization experiment, for instance, by applying dimensionality reduction to the learned representations and color-coding the data points by their class or attributes. However, as I am not deeply familiar with this specific area, I'll leave it to your judgment whether to include such an experiment. My primary concern is the novelty of the proposed method.

---

> > > > > ### Author Response · Authors · 2025-11-27
> > > > > **Novelty of the Hydra architecture**
> > > > >
> > > > > Thank you for the discussion. Your reviews earlier mentioned that our writing is clear and our results are superior. Hence, if the doubt is on novelty, we would like to offer a brief discussion because **Hydra is a state-of-the-art in multiple aspects, such as task structure (both regression and classification), scale (up to 64 networks), and time complexity**.
> > > > >
> > > > > The overall objective of the Hydra project is to design a transferable temporal graph model that can transfer effectively to unseen networks while answering important graph-level questions about the future. The majority of TGNN architectures are designed for the useful link prediction task, which mainly requires spatial and temporal components. The spatial component of the GNN models local interactions of nodes, and the temporal component updates the embeddings over time to model the temporal evolution of the graph structure. Therefore, TGNNs are effective for local tasks such as link prediction but lack in the understanding of global geometric information on the temporal graph. This is because the expressiveness of a GNN is bounded by the Weisfeiler-Leman graph isomorphism test, as shown in [1].
> > > > >
> > > > > To include the global graph information, our Hydra architecture utilizes an approximation of the graph Laplacian spectrum, including snapshots of how the distribution of the eigenvalues evolves over time. As shown empirically in the paper (through ablation studies in Table 3), Hydra achieves strong performance as we combine the best of spectral, spatial, and temporal components. In addition, spectral embeddings have rarely been used in temporal graphs, and we are one of the first works to incorporate spectral information about the global graph topology in the architecture. This is because traditional methods of computing eigenvalues and eigenvectors from SVD are computationally expensive and don't scale to large graphs (especially considering you have a whole sequence of them in temporal graphs). In contrast, the density of states approximation we use scales linearly to the number of edges and can be computed efficiently on large graphs.
> > > > >
> > > > > Hence, Hydra brings both scale and a type novelty: it is the first temporal graph model that can track global spectral patterns over time at practical cost and fuse them with spatial and temporal updates in one architecture. This gives Hydra a capability that prior TGNNs do not have, namely learning transferable representations that reflect how the entire graph evolves, not just local neighborhoods.
> > > > >
> > > > > [1] Xu, K., Hu, W., Leskovec, J., & Jegelka, S. How Powerful are Graph Neural Networks?. In the International Conference on Learning Representations.

---

> > > > > > ### Comment · Reviewer_Bzpw · 2025-11-27
> > > > > > **Response to authors**
> > > > > >
> > > > > > Thank you for your further clarification on novelty. Since it addresses my concerns, I raise my score to 6.

---

### Official Review · Reviewer_oMJR · 2025-11-01

**Soundness:** 3
**Presentation:** 3
**Contribution:** 3
**Rating:** 6
**Confidence:** 2

**Summary:**

1. The paper propose Hydra, an architecture capable of handling multiple temporal graph prediction tasks without retraining.
2. Hydra incorporates Laplacian descriptors into TGNNs for graph-level prediction.
3. The paper compares Hydra with previous methods for single-model/task prediction using multi-task classification.

**Strengths:**

1. The submission provides with the implementation and code.
2. Hydra is the first model capable of handling multiple temporal graph prediction tasks without retraining.

**Weaknesses:**

1. The experiments do not report the the performance drop brought by Hydra. If the per-task AUC/MAE of Hydra is much lower than training models per task, this proposed new multi-task learning paradigm seems to have more disadvantages than advantages.
2. Lack of citations to relevant papers [1, 2].

[1] Temporal graph benchmark for machine learning on temporal graphs

[2] Benchtemp: A general benchmark for evaluating temporal graph neural networks

**Questions:**

If we directly apply the modifications required for the temporal graphs to multi-task GNN architectures, how does this combination compare to Hydra?

---

> ### Author Response · Authors · 2025-11-21
> **Rebuttal by Authors (Part1)**
>
> We thank the reviewer for the time and effort spent evaluating our paper. We appreciate the insightful feedback and address each comment in detail below. We added revisions in blue.
>
>  **W1.Evidence on Hydra’s Multi-task Advantages**
> > The experiments do not report the the performance drop brought by Hydra. If the per-task AUC/MAE of Hydra is much lower than training models per task, this proposed new multi-task learning paradigm seems to have more disadvantages than advantages.
>
> **Response:** We appreciate the reviewer’s comment regarding potential performance drops in a multi-task setting. We do not see such a drop, instead we observe that multi-task training actually benefits model training; we will detail the performance increase i) moving from a single task model to Hydra, and ii) from single task Hydra to multi task Hydra.
>
> **First, multi-task Hydra outperforms single-task competitor models, such as HTGN**. In our experiments, we do not observe a systematic degradation when moving from per-task training to Hydra. As summarized in Tables 1 and 2 (with full per-token results in Section I of the appendix), Hydra achieves the most rank-1 results across almost all tasks and networks, despite being evaluated in a strictly more challenging regime: zero-shot transfer to unseen networks. In contrast, the single-task baselines are trained directly on each target network.
>
> Quantitatively, Hydra either matches or exceeds the best single-task model on the majority of datasets, and when it is not the top performer, the per-task gaps are small. Importantly, these results are obtained via zero-shot transfer on the test networks. This consistency across heterogeneous, unseen networks indicates that the shared representation learned by Hydra does not suffer from a multi-task penalty; instead, it generalizes effectively while providing the practical advantages of a single, reusable model.
>
> **Second, Hydra performance does not deteriorate but improves when more tasks are used in training**. We trained Hydra under three settings to evaluate how multi-task training affects regression performance on the edge-count prediction task:
> - **One-task Hydra:** trained only on the edge-count regression task and evaluated on the same task.
> - **Two-task Hydra:** trained on edge count plus a second regression task (new node count).
> - **Three-task (full) Hydra:** trained jointly on all three tasks in the model.
>
> The table below reports the MAE of the edge-count prediction across all datasets for each training configuration.
> This experiment shows that adding more tasks consistently improves regression performance on edge-count prediction. The two-task and especially the full three-task Hydra benefit from additional supervisory signals, which help the shared temporal representation capture richer structural dynamics. As seen in the table, the three-task model achieves the lowest MAE on most datasets and the best overall average, indicating that multi-task learning strengthens the model rather than introducing interference. This confirms that Hydra’s design effectively leverages complementary tasks to enhance performance on individual objectives. We have added these results to Section 5 of the main paper.
> |Dataset|One Task|Two Task|Three Task|
> |-------|--------|--------|----------|
> |MIR|0.057|0.034|**0.016**|
> |DOGE2.0|0.169|**0.150**|0.187|
> |MUTE|0.053|0.067|**0.021**|
> |EVERMOON|0.029|0.042|**0.017**|
> |DERC|0.024|0.035|**0.021**|
> |ADX|**0.023**|0.027|0.025|
> |HOICHI|0.071|0.068|**0.027**|
> |SDEX|0.060|**0.046**|0.095|
> |BAG|0.061|0.067|**0.041**|
> |XCN|0.037|**0.026**|0.062|
> |ETH2x-FLI|0.030|0.038|**0.023**|
> |stkAAVE|0.026|**0.021**|0.051|
> |GLM|0.092|**0.083**|0.087|
> |QOM|0.025|**0.023**|0.036|
> |WOJAK|0.025|0.039|**0.010**|
> |DINO|0.031|0.041|**0.017**|
> |Metis|0.047|0.054|**0.034**|
> |REPv2|0.103|**0.102**|0.111|
> |TRAC|0.029|0.039|**0.021**|
> |BEPRO|0.031|0.046|**0.011**|
> |**Average**|0.051|0.052|**0.046**|
> |**Best Count**|1|7|**12**|
>
> ---
> **W2.Missing Citations**
> > Lack of citations to relevant papers [1, 2].
>
> **Response:** We thank the reviewer for pointing out the missing citations. These works are indeed closely related: they also study multi-task or multi-network learning on temporal graphs and provide important context for our setting. We have added both [1, 2] in the revised version and now discuss how Hydra differs from their architectures and transfer regimes in Section 2.

---

> ### Author Response · Authors · 2025-11-21
> **Rebuttal by Authors (Part2)**
>
> **Q1.Comparison to Direct Multi-Task Temporal GNN Extensions**
> > If we directly apply the modifications required for the temporal graphs to multi-task GNN architectures, how does this combination compare to Hydra?
>
> **Response** We appreciate the reviewer’s question. A direct extension of a standard multi-task GNN to temporal graphs would typically consist of a shared GNN backbone on each snapshot, followed by a recurrent module (for example, a GRU) and task-specific heads. This is a natural baseline, and Hydra can be viewed as a structured instance of this general template, with two key differences:
>
> 1. Temporal stability and long-range structure. Standard GNN architectures are highly effective at capturing local embeddings of nodes based on their k-hop neighborhoods. However, in our graph-level task, global representation of the graph is also highly beneficial. To obtain the best from local and global representations, Hydra combines a spectral path (density of states with spectral memory) and a spatial path (snapshot GNN plus pooling and a temporal GRU). Placing a recurrent layer on a static multi-task GNN will lack the global spectral features from Hydra thus potentially being less powerful.
>
>  2. Transfer across heterogeneous networks. Classic multi-task GNNs assume a single underlying graph, so the shared representation is tied to one topology. Our setting requires multi-network transfer: the model is trained on dozens of token networks with different sizes, densities, and temporal behaviors and evaluated zero-shot on unseen tokens. Hydra is explicitly built for this regime by using graph-level representations and shared spectral statistics that are comparable across networks.
>
>
> In addition, Hydra jointly handles both multi-task regression and classification objectives in this transfer setting, while existing multi-task GNNs are typically evaluated on single-graph, single-type tasks. We will clarify this connection to multi-task GNNs in the revised manuscript and describe how Hydra can be interpreted as a principled temporal, multi-network extension of that family of models.

---

> > ### Comment · Reviewer_oMJR · 2025-11-26
> >
> > Thank you for your reply. My concerns have been resolved.
> >
> > Since I am not an expert in this domain, I will retain my rating.

---

> > > ### Author Response · Authors · 2025-11-27
> > >
> > > Thank you for the discussion and for your acceptance review.

---

### Author Response · Authors · 2025-11-30
**Official Author Comment for the Area Chairs**

Dear chairs,

We have revised the Hydra submission and addressed the reviewers’ comments with new experiments, clarifications, and additional analysis. In the updated PDF, responses are color-coded by reviewer. Below, we summarize the status of the main concerns.

>Reviewer oMJR (**rating 6**)

Asked whether multi-task Hydra retains per-task performance compared to separate models and whether adding tasks helps or hurts. They also asked how Hydra compares to a direct multi-task extension of standard temporal GNNs. We added a new experiment training Hydra in one task, two task, and three task settings on the edge count regression task. Three task Hydra achieves the best average MAE and the most per dataset wins, showing that multi task training strengthens rather than weakens individual tasks. We also clarified how Hydra’s spectral plus spatial trunk differs from naive multi task extensions. After the rebuttal, the reviewer considered the concerns resolved.

>Reviewer Bzpw (**rating 4 → 6**)

Raised concerns about novelty, the benefits of the trunk and head structure, the scope of multi-task settings, training time comparisons, and missing analysis on challenging tasks and regression ablations. We clarified that the novelty lies in combining spectral temporal fusion with multi-network, multi-task transfer, rather than introducing new primitives. Hydra is, to our knowledge, the first model that handles multi-task classification and regression on temporal graphs with zero-shot transfer to unseen networks. We added regression ablations, extended the multi-task study to one, two, and three task Hydra, and provided a deeper analysis of failure modes, such as influential node count and the scaling behavior in Figure 3. We also explained how the trunk amortizes temporal computation across tasks. After further discussion, the reviewer raised the score.

>Reviewer mbwn (**rating 4**)

Agreed that the paper is clear and the multi-network results are strong, but asked for stronger motivation, raised concerns about baselines and the relation to MiNT, and noted that training classification and regression separately limits scope. We expanded the motivation and Appendix E to ground the setting in real applications, particularly blockchain transaction networks where many token graphs must be monitored for global risk indicators such as fragmentation and edge surges. We clarified that similar global temporal properties already appear in GraphPulse and MiNT, and that Hydra is designed as a single reusable model for unseen tokens. We also clarified the architectural and training differences from MiNT and explained why the trunk and head design naturally supports future joint classification plus regression training.

>Reviewer xXNB (**rating 4 → 6**)

Focused on novelty in the multi task paradigm and the complexity and positional encoding role of the density of states module. We clarified that the contribution is not the multi-task interface but the spectral-temporal trunk tailored to multi-network graph level prediction, which must capture global structural evolution and support multiple tasks on a single representation. We highlighted the performance gains over strong temporal baselines as evidence that simply adding heads is not enough. On complexity, we explained that density of states uses the kernel polynomial method, giving linear complexity in the number of edges rather than cubic in the number of nodes. We added empirical timing up to tens of millions of edges and clarified the distinction from eigenvector-based encodings. After these clarifications, the reviewer increased the score.

Across all reviews, no one questioned the methods or experiments. Two reviewers raised their scores, and the remaining concerns focus on framing novelty and future extensions, such as mixed classification plus regression, additional baselines, and representation visualizations. We believe the added experiments, expanded motivation, and clarifications substantially strengthen the case for Hydra as a concrete step toward adaptable foundation models for temporal graphs.


Best Regards,

The Authors

---

### Meta-Review · Area_Chair_Jj5M · 2025-12-22

**Summary:**

The paper proposes Hydra, a multi-task temporal graph model that combines local temporal GNN features with a spectral module to capture global connectivity, aiming to improve transfer and generalization across tasks and networks. Empirically, Hydra improves multi-task classification and regression performance over competitive baselines while training up to 22× faster than temporal transfer models.

Reviewers raised concerns about limited technical novelty, unclear motivation, insufficient justification of the practical relevance of multi-task global property prediction on temporal graphs, and unconvincing evaluation benchmarks. The rebuttal partially addresses these issues by clarifying the novelty, adding baseline comparisons and ablations. However, the central concern, which is whether the proposed problem setting is well-motivated and practically meaningful, remains insufficiently resolved. A stronger way to address this would be to include concrete real-world use cases in the evaluation, which would require a substantial update to the experimental section.

**Reviewer Concerns:**

The rebuttal clarifies the technical novelty and strengthens the empirical evaluation by adding baseline comparisons and ablations. However, concerns about the motivation and practical relevance of the proposed problem setting remain insufficiently addressed.

**Reviewer Scores:**

Two reviewers indicated they would increase their scores. However, the more critical review is likely to remain unchanged, since the key issues, particularly around motivation and practical relevance,  are not adequately addressed.

---

### Decision · Program_Chairs · 2026-01-26

Reject